# The conserved protein Seb1 drives transcription termination by binding RNA polymerase II and nascent RNA

Sina Wittmann[1], Max Renner[2], Beth R. Watts[1], Oliver Adams[1], Miles Huseyin[1], Carlo Baejen[3], Kamel El Omari[4], Cornelia Kilchert[1], Dong-Hyuk Heo[1], Tea Kecman[1], Patrick Cramer[3], Jonathan M. Grimes[2,4] & Lidia Vasiljeva[1]

Termination of RNA polymerase II (Pol II) transcription is an important step in the transcription cycle, which involves the dislodgement of polymerase from DNA, leading to release of a functional transcript. Recent studies have identified the key players required for this process and showed that a common feature of these proteins is a conserved domain that interacts with the phosphorylated C-terminus of Pol II (CTD-interacting domain, CID). However, the mechanism by which transcription termination is achieved is not understood. Using genome-wide methods, here we show that the fission yeast CID-protein Seb1 is essential for termination of protein-coding and non-coding genes through interaction with S2-phosphorylated Pol II and nascent RNA. Furthermore, we present the crystal structures of the Seb1 CTD- and RNA-binding modules. Unexpectedly, the latter reveals an intertwined two-domain arrangement of a canonical RRM and second domain. These results provide important insights into the mechanism underlying eukaryotic transcription termination.

[1] Department of Biochemistry, University of Oxford, Oxford OX1 3QU, UK. [2] Division of Structural Biology, Wellcome Trust Centre for Human Genetics, University of Oxford, Oxford OX3 7BN, UK. [3] Department of Molecular Biology, Max Planck Institute for Biophysical Chemistry, 37077 Göttingen, Germany. [4] Diamond Light Source Ltd, Harwell Science & Innovation Campus, Didcot OX11 0DE, UK. Correspondence and requests for materials should be addressed to L.V. (email: lidia.vasilieva@bioch.ox.ac.uk).

Termination of RNA polymerase II (Pol II) transcription is a fundamental but poorly understood step in gene expression. Timely and efficient termination is essential for the production of functional mRNAs and influences pre-mRNA processing events including the choice of polyadenylation site (PAS). Therefore, deregulated transcription termination has dramatic impacts on the localization, stability and coding potential of transcripts. Moreover, failure to terminate can interfere with the function of downstream promoters[1–3]. Although key termination factors have been identified, it is still largely unknown how they promote transcription termination. Many of these factors including Nrd1, Pcf11 and Rtt103 directly interact with the phosphorylated C-terminal domain (CTD) of Pol II via their conserved CTD-interacting domain (CID) (Fig. 1a). The CTD itself comprises conserved heptad repeats ($Y^1S^2P^3T^4S^5P^6S^7$) and five of the residues can be phosphorylated (S2P, S5P, S7P, Y1P and T4P) during the transcription cycle[4–6]. Termination also depends on the recognition of the PAS at the 3′ end of nascent transcripts by the multiprotein cleavage and polyadenylation factor (CPF) and the cleavage factor IA (refs 7–12). Endonucleolytic cleavage at the PAS provides an entry point for a 5′-3′ exonuclease (Rat1 in budding yeast, Saccharomyces cerevisiae (S. cerevisiae)/Xrn2 in human/Dhp1 in fission yeast, Schizosaccharomyces pombe (S. pombe))[3,4,13–15] which degrades nascent RNA and according to the so-called 'torpedo model' of termination is necessary for dislodging Pol II from DNA (reviewed in ref. 16). However, recent studies demonstrated that cleavage at the PAS, as well as the presence of Xrn2, may not be absolutely required for transcription termination[17–19] suggesting that an additional mechanism is likely to contribute. Indeed, non-coding transcripts lacking PASs terminate in a Rat1-independent manner in budding yeast[14,20]. Instead, these transcripts rely on the CID termination factors Nrd1 and Pcf11 (Fig. 1a)[14,21,22]. Nrd1 interacts with initiating S5P-Pol II (ref. 13) and terminates short (≤1 kb) transcripts[14,23]. Nrd1 also contains an RNA recognition motif (RRM)[24,25] and interacts with another RRM protein, Nab3, and the RNA helicase Sen1 to form the NNS (Nrd1-Nab3-Sen1) complex[26]. Nrd1 and Nab3 recognize UGUAA/G and UCUUG motifs, respectively, which constitute Nrd1-dependent terminators[24,27–29]. Nrd1 also interacts with components of the nuclear exosome complex[26] and facilitates processing of small nuclear and small nucleolar (sn and sno) RNAs as well as degradation of unstable non-coding transcripts[30]. In contrast to Nrd1, the CID-protein Pcf11 interacts with S2P-Pol II (refs 31,32) and is important for termination of both PAS-dependent (protein-coding) as well as PAS-independent (non-coding) genes[14]. The fact that Pcf11 functions universally in termination appears to be conserved in all eukaryotes. Both Nrd1 and Pcf11 are essential in S. cerevisiae, reflecting their critical and complementary roles in termination.

In general, CID-RRM proteins are highly conserved among eukaryotes (Fig. 1a). However, despite the high degree of homology between Nrd1 and the human CID-RRM proteins Scaf4 and Scaf8, the existence of two distinct termination pathways for non-coding and protein-coding transcripts does not appear to be conserved in humans[33].

Here we show that, like Scaf8 (ref. 34), the essential CID-RRM protein Seb1 from fission yeast shows preference for S2P-Pol II both in vitro and in vivo. In agreement with a recent study[35], we show that Seb1 co-purifies with components of the CPF machinery and is needed for efficient pre-mRNA cleavage. Using PAR-CLIP (Photoactivatable Ribonucleoside-Enhanced Crosslinking and Immunoprecipitation) and ChIP-Seq (chromatin immunoprecipitation-sequencing), we demonstrate that Seb1 is recruited to the 3′ ends of genes. In contrast to Nrd1, Seb1 functions in terminating all classes of Pol II-transcribed genes genome-wide. To further dissect the molecular underpinnings of RNA and Pol II recognition by Seb1, we performed structural analyses utilizing X-ray crystallography and small angle X-ray scattering (SAXS). We present high-resolution structures of the CID and RRM domain of Seb1. Notably, our 1.0 Å structure of the RRM domain reveals an unusual arrangement of a classical RRM domain interwoven with a compact second domain which are both needed for RNA binding and recognition. To the best of our knowledge, this uncommon configuration of an RNA-binding module has not been observed before. Based on these structures, Seb1 point mutations were designed that abolish binding either to S2P-CTD or nascent RNA. This results in global deregulation of transcription due to severe RNA processing and transcription termination defects. Using a multidisciplinary approach, our study demonstrates that conserved CID-RRM proteins play a key role in 3′ end formation of Pol II transcripts.

## Results

**Seb1 interacts with the CPF and binds at the 3′ end of genes.** Seb1 contains a CID which is conserved specifically in eukaryotic termination factors (Fig. 1a). Therefore, to investigate whether it is also involved in transcription termination we purified Seb1 in complex with other proteins and identified them by mass spectrometry (Supplementary Fig. 1a). In addition to expected interactors such as subunits of Pol II, several CPF components co-purify with Seb1 suggesting, in agreement with a recent publication[35], that it is involved in mRNA 3′ end formation.

We also mapped Seb1-RNA interaction sites transcriptome-wide using PAR-CLIP[25,29]. A motif search (±25 nucleotides (nt) around crosslinking sites) showed an enrichment of Seb1 over UGUA (Fig. 1b), resembling the Nrd1 binding motif. Consistent with our data, a similar Seb1 motif (A(U)GUA) has been identified using CRAC (crosslinking and cDNA analysis)[35]. Interestingly, Seb1 binding is preferentially observed downstream of PASs where the UGUA motif is also specifically enriched (annotations used from ref. 36, Fig. 1c). Previous bioinformatic analyses had also identified this motif as a putative element involved in 3′ end formation of protein-coding genes in fission yeast[37]. Together these data suggest that this element constitutes a functionally important transcription termination site.

To test whether Seb1 is recruited co-transcriptionally, we performed Seb1 ChIP-Seq. Indeed, Seb1 is found at the 3′ end of transcription units (Fig. 1c, Supplementary Fig. 1b and c) albeit further downstream (∼160 nt after the PAS) compared to PAR-CLIP (∼80 nt after the PAS). This suggests that Seb1 is associated with the transcription machinery after synthesis of the transcription termination site. In contrast, the frequent occurrence of Seb1 crosslinks to the 5′ end of transcripts is not reflected in a corresponding enrichment of the protein on chromatin by ChIP (Fig. 1c). It is possible that interactions between Seb1 and the 5′ end of transcripts are transient or occur post-transcriptionally. Although Seb1 motifs are frequently found within gene bodies, Seb1 is depleted from these regions, suggesting that binding to RNA is not the only determinant of Seb1 recruitment.

On non-coding genes, Seb1 shows stronger binding to the 5′ end but less recruitment to the 3′ end as compared to protein-coding genes (Fig. 1c and Supplementary Fig. 1d). Seb1 also crosslinks on TSS- and PAS proximal antisense transcripts (Supplementary Fig. 1e) suggesting that it may promote termination of cryptic transcripts that are initiated from open chromatin at promoters and terminators.

Of the 4,228 protein-coding and non-coding genes that were included in the analysis, ∼63% show recruitment of Seb1 to the

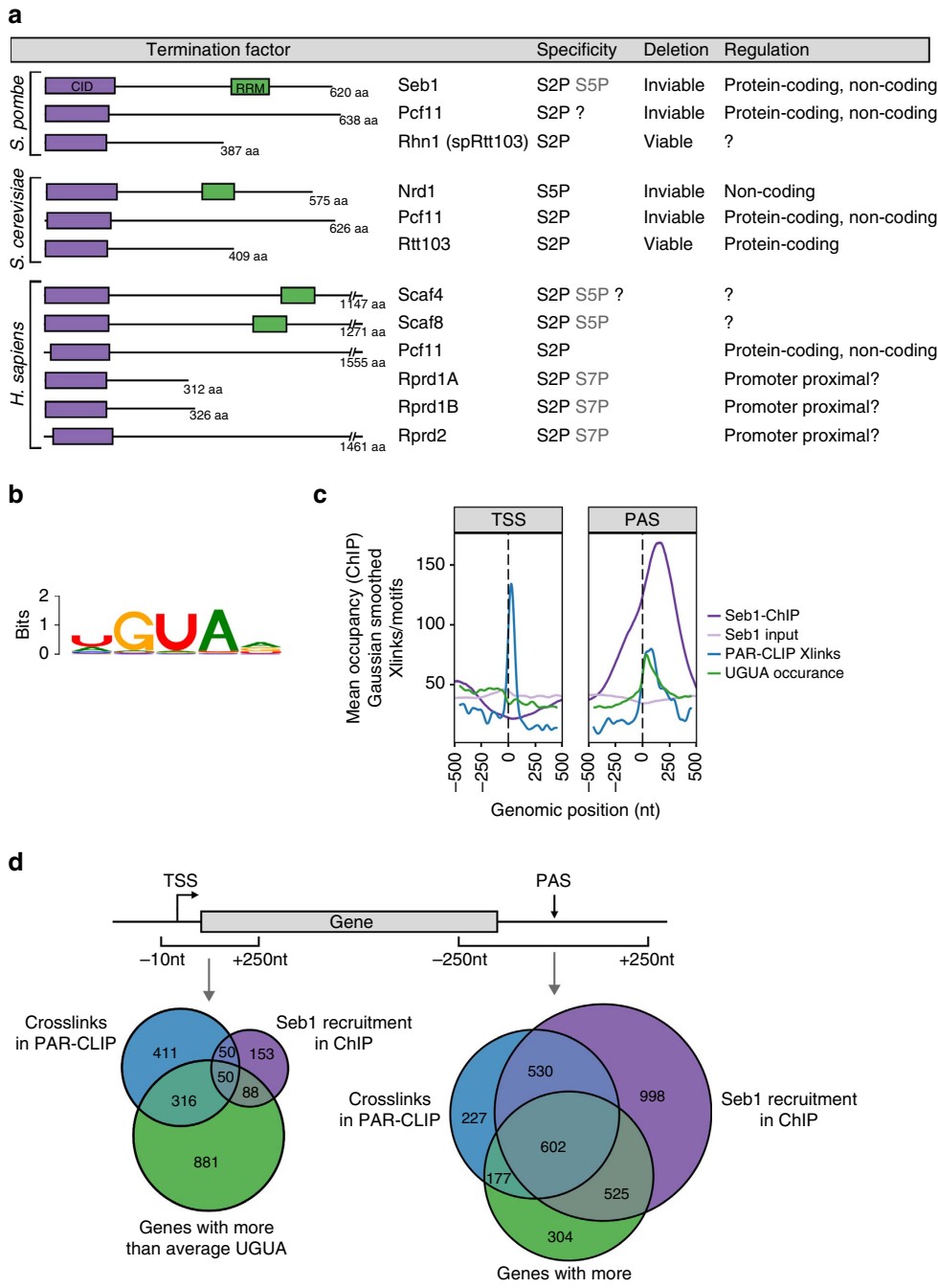

**Figure 1 | Seb1 localizes to the 3′ end of genes and interacts with the CPF.** (**a**) Comparative overview of homologous CID-containing proteins from *S. pombe*, *S. cerevisiae* and *H. sapiens*. CTD specificities are based on this study, published data or inferred from sequence alignments (in Supplementary Fig. 2d). (**b**) The Seb1 binding motif as determined by PAR-CLIP is shown. The motif occurrence is 42.74% in a window of ± 25 nt around the crosslinked site (XXmotif E-value: $5.34 \times 10^{-55}$). (**c**) Averaged occupancy profiles of Seb1 and input from ChIP, PAR-CLIP crosslinks and occurrence of the Seb1 binding motif UGUA, normalized to transcript levels are shown. The profiles are aligned to the TSS and PAS as indicated. Genes that have less than a 250 nt distance to their downstream gene or that are shorter than 500 nt were excluded from the analysis ($n = 4,228$). The PAR-CLIP and motif profiles were smoothed using a Gaussian smoothing function and adjusted to bring to scale with the ChIP-seq profile. (**d**) Overlap between Seb1 binding in PAR-CLIP and ChIP as well as motif occurrence are shown as Venn diagrams. Presence of crosslinks, ChIP peak summits, or a higher than average motif occurrence in a window of 10 nt before to 250 nt after the TSS are shown on the left, and 250 nt around the PAS on the right. The same subset of genes was used as in **c**.

PAS ± 250 nt by ChIP-Seq and about 36% by PAR-CLIP (Fig. 1d and Supplementary Data 1). Overall, 74% of the genes that show crosslinks by PAR-CLIP also show Seb1 enrichment by ChIP-Seq, suggesting that Seb1 interacts with RNA co-transcriptionally. In addition, ∼81% of genes that show a high frequency of UGUA occurrences in the PAS region recruit Seb1 by either ChIP-Seq or PAR-CLIP, underscoring the importance of the motif for Seb1 binding. In contrast, at the TSS (10 nt upstream to 250 nt downstream), there are more binding events detected by PAR-CLIP than by ChIP-Seq. Surprisingly, the presence

of UGUA in this region seems to be less important for Seb1 recruitment (Fig. 1c). Furthermore, less than half of the binding events at the 5′ end show simultaneous recruitment at the 3′ region (Supplementary Fig. 1f), suggesting that the two events are independent of one another.

**Both domains of Seb1 are essential.** Seb1 has two ordered domains, a CID at the N-terminus and an RRM-containing RNA-binding region closer to the C-terminus. The rest of the protein is intrinsically disordered. To examine which parts of the protein are functionally important, we constructed seven different mutants in which consecutive regions are deleted (Fig. 2a). As some of these strains are expected to be inviable, we constitutively expressed a wild-type (WT) copy of Seb1 in parallel using the thiamine-repressible $nmt1$ promoter. After 24 h in thiamine-containing medium, most of WT-Seb1 is depleted and all seven truncated proteins are stably expressed (Supplementary Fig. 2a). Under those conditions, strains that have the CID or RRM domain deleted are inviable suggesting that binding to both Pol II and RNA is important (Fig. 2a). In addition, the regions directly following the domains are also essential (Fig. 2a). Deletion of the region after the CID might interfere with folding or it could be engaged in protein–protein interactions, as was shown for Nrd1 (ref. 13). The region after the RRM was proposed to contribute to RNA binding in Nrd1 (ref. 38) and our data suggest that this is also the case in Seb1 (see below).

**Specificity of Seb1 for phosphorylated Pol II.** We next asked with which form of phosphorylated Pol II Seb1 can interact. Using fluorescence anisotropy (FA), we studied the binding of recombinantly expressed $CID_{1–152}$ to FAM-labelled, differently modified two-repeat CTD peptides (Fig. 2b). The highest affinity was observed to S2P, followed by S5P, S7P and unphosphorylated peptide (Fig. 2b and Table 1). Consistently, Seb1 interacts with S2P- and S5P-Pol II $in$ $vivo$ (Supplementary Fig. 2b) and Seb1 purified from yeast can also bind to four-repeat S2P- and S5P-, but not S7P- or unphosphorylated peptides (Fig. 2c). As S2P peaks at the 3′ end of transcription units, these data are in agreement with the observed localization of Seb1 on chromatin. This is very different from Nrd1 which is recruited at the beginning of the transcription cycle via binding to S5P-CTD (ref. 13) but similar to human Scaf8 which was demonstrated to bind S2P-CTD peptides[34]. Since S2P levels are higher at the end of long genes compared to short genes, we wanted to know if Seb1 binding correlates with gene length. Indeed, genes that are bound by Seb1 in ChIP-Seq, but not PAR-CLIP, are significantly longer than genes that are not (Supplementary Fig. 2c). This indicates that S2P might play a role in recruiting Seb1 to sites of transcription but is unlikely to be important for RNA binding $per$ $se$.

Next, we wanted to investigate the functional consequences of disrupting Seb1–Pol II interactions. We therefore attempted to identify amino acid (aa) mutations that specifically abolish binding to either S2P or S5P-CTD. Thus, we crystallized the $Seb1-CID_{1–152}$ (Table 2 and Supplementary Figs 3a–d, 2d) and compared the structure with other CID structures in complex with CTD peptides to determine whether interacting aa are topologically conserved.

Binding to S2P-CTD is known to be mediated by a basic aa (Arg or Lys), which directly contacts the phosphate moiety (R108 of Rtt103, Fig. 2e). In addition, another basic aa nearby (K105) directly interacts with the phosphate in some states of the Rtt103 NMR structure (Fig. 2e)[39]. Both aa are conserved in Seb1 (K124 and K121, respectively), as well as Scaf4 and

Scaf8 (Fig. 2d and Supplementary Fig. 3e), but not in Nrd1 which has low affinity to S2P-CTD (ref. 13).

The sole CID-protein known to prefer S5P-CTD is Nrd1 (ref. 13) whose crystal structure shows a Ser and Arg directly contacting the phosphate group (Fig. 2f)[40]. The former residue is conserved in all known CID proteins (S22 in Seb1, Fig. 2d); however, only Seb1, Scaf4 and Scaf8 have the latter aa conserved (K25 in Seb1, Fig. 2d and Supplementary Fig. 3e). Also, several aa make phosphate-independent contacts with the CTD. Most notably, a Tyr (Y64 in Seb1, Fig. 2d and Supplementary Fig. 3e) forms an aromatic (π–π) interaction with Y1 of the CTD in all published CTD-CID structures (Y62 in Fig. 2e and Y67 in Fig. 2f) and a conserved Asp (Supplementary Fig. 3e) was shown to be important for Nrd1 function[40]. We therefore introduced all aforementioned mutations into full-length Seb1 and some selected mutations into the $Seb1-CID_{1–152}$ (Supplementary Fig. 4a and b). The proteins were recombinantly expressed and purified, and binding to CTD peptides was tested. In the case of the full-length protein, four-repeat CTD peptides immobilized on streptavidin beads were used. Binding of the $Seb1-CID_{1-152}$ was assessed quantitatively using FA and two-repeat peptides (Fig. 2g–i).

As expected, changing the charge of the two basic aa K121 and K124 predicted to contact S2P resulted in severely reduced binding (Supplementary Fig. 4a, lanes 6 and 8 and Fig. 2g). When these two residues are mutated to Ala, binding to S2P-CTD can still be observed (Supplementary Fig. 4a, lanes 7 and 9) suggesting that both aa may contribute to the S2P interaction.

The S22D and K25E single mutations had little effect on CTD binding (Supplementary Fig. 4a, lanes 2 and 3); however, S22D-K25E combined resulted in reduced binding to S5P-CTD (Supplementary Fig. 4a, lane 10 and Fig. 2h). Strikingly, the triple mutation S22D-K25E-K124E most severely affects binding to S2P and S5P peptides, more than any of the individual mutations (Supplementary Fig. 4a, lane 11; Fig. 2g and h). This suggests that the introduced negative charges somewhat destabilize peptide interactions independently of their phosphorylation state.

Y64 and D67 were predicted to be involved in phosphorylation-independent interactions and to therefore affect binding to all types of CTD. Indeed, Y64K reduces binding to all phosphorylated and unphosphorylated peptides; however, it has a stronger effect on S2P than S5P interaction (Supplementary Fig. 4a, lane 4 and Fig. 2g–i). This could be explained by the spatial proximity to the S2P binding pocket. D67M, on the other hand, had no effect on binding (Supplementary Fig. 4a, lane 5).

To evaluate how binding to different Pol II phospho-isoforms contributes to the function of Seb1 $in$ $vivo$, we introduced S22D, S22D-K25E, Y64K, K124E, K121E and S22D-K25E-K124E mutations into yeast cells alongside the repressible Seb1-WT copy. When only mutated Seb1 is expressed (Supplementary Fig. 4c), the S22D-K25E, K121E and K124E mutations, which only moderately affect S2P and S5P binding, show a mild growth phenotype in comparison to WT (Fig. 2j). In contrast, the Y64K and S22D-K25E-K124E mutants are nearly lethal. Despite Seb1 having its highest affinity to S2P-CTD, this demonstrates that disrupting interactions with S2P-CTD alone is not sufficient to observe the full phenotype, and that interactions with S5P-CTD are also required for proper functioning of Seb1.

**Structural and functional analyses of Seb1 RNA binding.** Our observation that the RRM domain is essential for cell viability suggests that binding to RNA is important for Seb1 function. Here, too, we adopted a structure-based approach. The Seb1-RRM domain is unusual in that it is flanked by additional regions which are conserved in its homologues

Nrd1, Scaf4 and Scaf8 (Supplementary Fig. 5a). Furthermore, the NMR structure of the Nrd1-RRM showed a largely unstructured helix-loop bundle lying C-terminally to a classical RRM fold[38].

We expressed Seb1-RRM$_{388-540}$ and crystallized the purified domain (Supplementary Fig. 5b-e), yielding diffraction data up to 1.0 Å. The structure was phased using sulfur single wavelength anomalous dispersion (S-SAD, Table 2). Surprisingly,

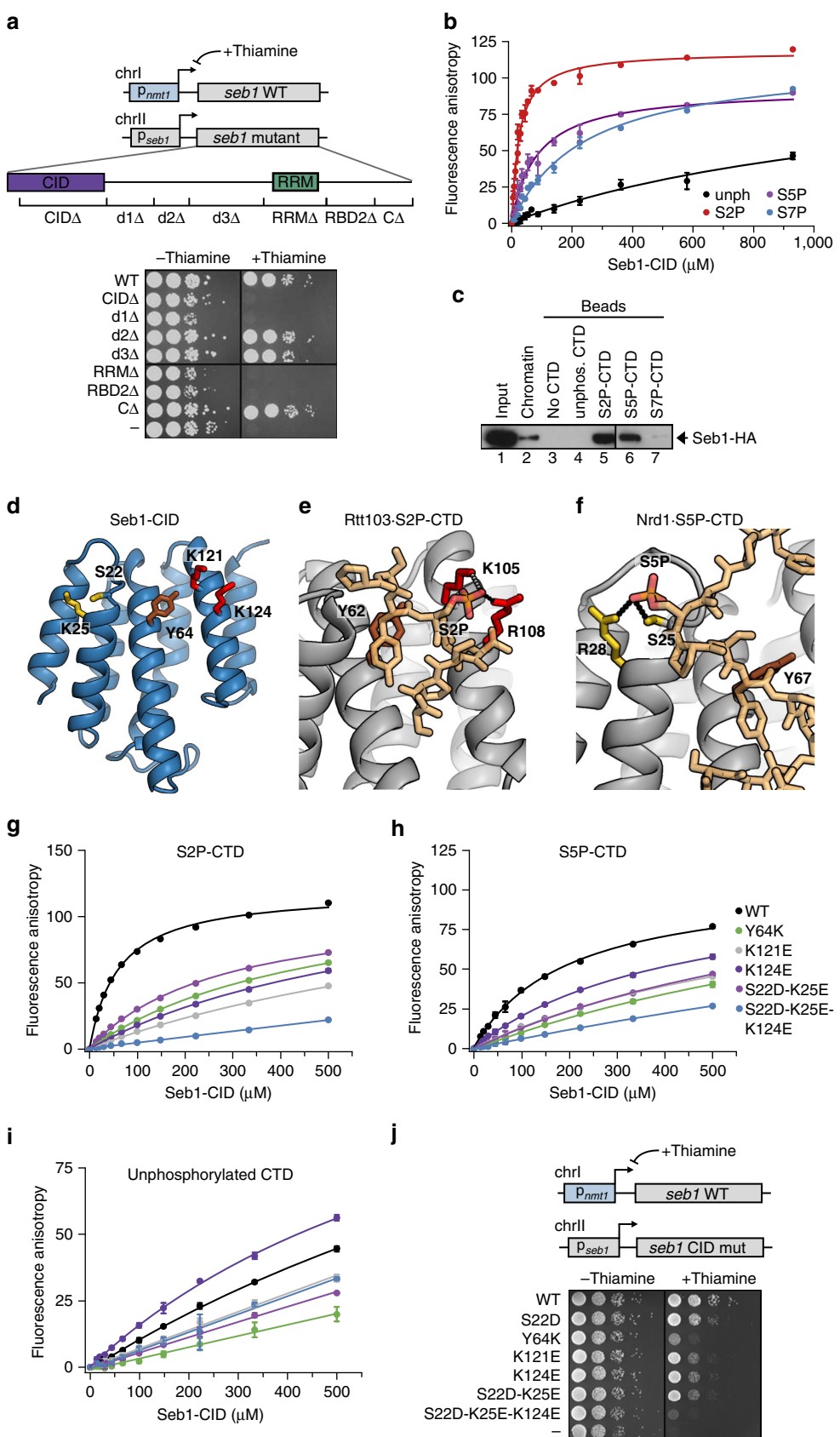

**Figure 2 | The Seb1-CID preferentially binds to S2P-CTD and is required for viability. (a)** Spot test showing growth of Seb1 deletion mutants on media containing or lacking thiamine ( + and – thiamine, respectively). The strains carry a thiamine-repressible WT and a mutated Seb1 copy under control of the endogenous promoter. **(b)** Binding of Seb1-CID$_{1-152}$ to the FAM-tagged two-repeat non-phosphorylated or phosphorylated CTD peptides measured by FA. Error bars show the standard deviation of at least three independent repeats. **(c)** Binding assays of IgG purified Seb1-HA-TAP to biotinylated non-phosphorylated or phosphorylated four-repeat CTD peptides immobilized on streptavidin beads analysed by western blot using α-HA antibody. **(d)** Crystal structure of the Seb1-CID$_{1-152}$. Amino acids that were later changed by mutagenesis *in vivo* are depicted as sticks. Yellow amino acids are involved in S5P recognition, red amino acids are important for S2P binding and Y64 (brown) interacts with the CTD independently of any phosphate moieties. **(e)** Structure of *S. cerevisiae* Rtt103 in complex with S2P-CTD (PDBID 2L0I). Topologically conserved residues, which are also found in Seb1 at equivalent positions (compare with **d**) and are involved in S2P-CTD recognition, are shown in red. R108 contacts the phosphate on S2 in most states of the NMR ensemble (dotted black line) while K105 binds only in some (dotted grey line). A key conserved Tyr is shown in brown. **(f)** Structure of *S. cerevisiae* Nrd1 in complex with S5P-CTD (PDBID 2LO6). Relevant amino acids located at equivalent positions to those shown for the Seb1-CID are coloured as in **d**. Residues contacting S5P are conserved in Seb1 (compare with **d**) and shown in yellow. **(g)** Binding of WT and mutated Seb1-CID$_{1-152}$ (as indicated) to S2-phosphorylated FAM-tagged double-repeat CTD peptides measured by FA. Error bars show the standard deviation of three technical replicates. **(h)** Same as **g** but binding to S5-phosphorylated CTD peptides was measured. **(i)** Same as **g** but binding to non-phosphorylated CTD peptides was measured. **(j)** Spot test showing the effect of the indicated Seb1-CID point mutations on cell growth on media containing or lacking thiamine as in **a**.

residues lying both N- and C-terminally of the canonical RRM domain (Supplementary Fig. 5a) fold together to yield a compact additional domain (Fig. 3a and b). The domain (denoted domain 2) contains two η-helices as well as four β-strands which form a β-sheet packed against one α-helix. To our knowledge, no other deposited RRM structure shows this unusual arrangement of interweaved distinct regions of primary sequence folding together to form two adjacent domains. A structural homology search with the Dali server[41] using only domain 2 yielded no significant hits.

Because the Nrd1 NMR structure is overall very different from the Seb1 crystal structure (with more loops and unfolded regions (Supplementary Fig. 6a)), we wanted to understand how the Seb1-RRM domain behaves in solution. We therefore collected SAXS data on the purified RRM$_{388-540}$ domain (Supplementary Fig. 6b and c). Comparison of the experimental data in solution and a theoretical curve obtained from the crystal structure shows excellent correlation (Fig. 3c), indicating that the crystal structure represents the solution conformation. Furthermore, an *ab initio* bead model calculated from the SAXS data closely fits the shape of the crystal structure (Supplementary Fig. 6d). Because the Nrd1-RRM structure shows a high degree of disorder and flexibility, which is in stark contrast to the Seb1-RRM$_{388-540}$ structure, we assessed the flexibility of the domain in solution (Fig. 3d). Dimensionless Kratky analysis indicates that Seb1-RRM$_{388-540}$ is highly ordered in solution, being only marginally less folded than the highly inflexible lysozyme standard. Taken together, these data show that the crystal structure of the Seb1-RRM$_{388-540}$ closely reflects the conformation of the protein in solution.

The electrostatic surface of the domain shows clear patches of positive charge (Fig. 3b, shown in blue), constituting potential RNA interaction sites. To map the RNA binding surface, we introduced several point mutations into the RRM$_{388-540}$ and measured their effect on binding to an FAM-labelled 10 nt-RNA containing the Seb1 binding motif (Supplementary Fig. 6e, Fig. 3e and Table 1). Mutating residues F445A and R472E, which are located on the β-sheet that is formed by the canonical RRM, completely abolishes RNA binding (Fig. 3e and Supplementary Fig. 6f). Interestingly, the F487A mutation which is located in a loop of domain 2 near η2 severely reduces the affinity to RNA. This suggests that the RNA likely interacts with the canonical RRM via its β-sheet (which constitutes the classical binding region for RRMs), and that the second domain may fold down onto the RNA, allowing F487 to engage in base-stacking interactions with the RNA, thereby increasing the domain's affinity and, possibly, specificity.

In addition, mutating residues that affect the interaction between the two domains resulted in insolubility (Supplementary Fig. 5a). Taken together with the fact that domain 2 is essential (Fig. 2a), this supports the notion that both parts, the canonical RRM and domain 2, fold together to form one rigid body that interacts with RNA.

We chose two mutations to test the effect of RNA binding *in vivo*—one that abolishes the interaction completely (F445A), and one that has only a mild effect on binding (S492A, Fig. 3e). Both mutants produced stable Seb1 protein in yeast (Supplementary Fig. 6g). The F445A mutant is inviable while S492A has no effect on cell growth (Fig. 3f). This unambiguously demonstrates that not only the presence of the RRM domain is essential for cells, but so too is its ability to bind RNA.

**Seb1 is required at protein-coding and non-coding genes.** Next, we wanted to study how loss of RNA or CTD binding affects Seb1 recruitment to chromatin. We therefore performed Seb1 ChIP-qPCR on two protein-coding genes, *rps401* and *pho1* (Fig. 4a and Supplementary Fig. 7a). All severely growth-impaired mutants (Y64K, S22D-K25E-K124E and F445A) show a strong decrease in recruitment to both genes. This is not the case for the tested mutants which display milder phenotypes. Analysis of the Pol II profile on *rps401* reveals noticeable downregulation of transcription. We therefore normalized Seb1 to Pol II levels (Fig. 4b) which, interestingly, shows that only the RRM mutant F445A showed loss of Seb1 recruitment. This suggests that binding of Seb1 to Pol II is not needed for recruitment. Furthermore, while Pol II levels drop past the PAS in WT, all mutants tested except S492A show little decrease in polymerase, suggestive of a failure to terminate transcription (Fig. 4b and Supplementary Fig. 7b).

We then asked how the impairment of Seb1 function affects the fission yeast transcriptome. Therefore, we conducted RNA-Seq in Seb1-WT, S22D-K25E, Y64K, S22D-K25E-K124E, F445A and S492A cells using ribo-depleted, total RNA (Supplementary Fig. 7c). To assess overall effects on 3′ end formation, we looked at transcript levels in the window 250 nt ± PAS (Fig. 4c). A metagene plot of these data shows high levels of transcriptional read-through in all inviable mutants as indicated by higher numbers of reads past the PAS compared to WT. We also took a more quantitative approach to assess read-through levels. For this, reads in the region from the PAS to 50 nt downstream were counted, normalized to gene-body counts and subsequently to normalized post-PAS levels in WT (Fig. 4d and Supplementary Data 1). All mutants except S492A show significantly more read-through than WT, suggesting that Seb1

is indeed necessary for proper 3′ end formation or transcription termination and that interaction with both Pol II and RNA are required. It should be noted that due to the unstable nature of read-through transcripts RNA-Seq likely underestimates their occurrence.

The relatively high level of intergenic reads in the mutants makes normalization to library size, which is most commonly used, inappropriate. We therefore normalized to reads coming from the

highly expressed housekeeping gene *adh1*, which does not show major changes in transcript levels (Supplementary Fig. 7d). Even though *adh1* mRNA levels are not affected, read-through transcription is observed in F445A (Supplementary Fig. 8a). RT–qPCR confirms the presence of read-through transcripts observed on *adh1* and *pho1* (Supplementary Fig. 8a–d). Furthermore, northern blot and RNA-Seq analysis of *rps401* shows striking accumulation of longer 3′ extended RNA species in Y64K, S22D-K25E-K124E and F445A (Fig. 4e and Supplementary Fig. 8e). Accumulation of 3′ extended transcripts upon loss of Seb1 was also reported by a recent study[35]. We performed a global analysis of the cleavage efficiency in Seb1 mutants using published PAS annotations[42]. Interestingly, our analyses revealed reduced usage of all PASs in Y64K, S22D-K25E-K124E and F445A (Supplementary Fig. 8f). These data suggest that Seb1 is likely needed for efficient cleavage of the pre-mRNA rather than influencing PAS choice as previously proposed[35]. Because 3′ extended transcripts are often unstable, a decrease in cleavage efficiency is therefore expected to result in less RNA overall. Indeed, accumulation of 3′ extended species coincides with a drastic decrease in *rps401* transcripts levels (Fig. 4e, WT: 100%, S22D-K25E: 82%, Y64K: 34%, S22D-K25E-K124E: 30%, F445A: 43%). In conclusion, this suggests that recruitment of Seb1 to the 3′ end of genes is necessary for proper 3′ end processing and termination of Pol II.

The severity of the transcription defect observed in the mutants can potentially influence the expression of many transcripts genome-wide (Supplementary Fig. 9a). Read-through transcription can lead to transcription interference (for example, *SPCC297.06c* reading into *set7* or *SPNCRNA.1239* and *SPCC1223.14*). In the most extreme case, multiple genes seem to be connected without any obvious termination in-between (indicated by red bar). Overall, the deregulation of transcription termination as a result of these

**Table 1 | $K_d$ values of CID–CTD and RRM–RNA interactions.**

| Domain | Mutant | Ligand | $K_d$ |
|---|---|---|---|
| CID | WT | Double-repeat CTD | >1,000 µM |
| CID | WT | Double-repeat S2P | 23.4 ± 2.7 µM |
| CID | WT | Double-repeat S5P | 83.7 ± 12.6 µM |
| CID | WT | Double-repeat S7P | 241.8 ± 28.7 µM |
| RRM cleaved | WT | RNA: AUUAGUAAAA | 1.84 ± 0.63 µM |
| SUMO-RRM | WT | RNA: AUUAGUAAAA | 1.72 ± 0.06 µM |
| SUMO-RRM | R392E | RNA: AUUAGUAAAA | 3.33 ± 0.53 µM |
| SUMO-RRM | K402A | RNA: AUUAGUAAAA | 2.49 ± 0.33 µM |
| SUMO-RRM | Y404A | RNA: AUUAGUAAAA | unstable protein |
| SUMO-RRM | T407A | RNA: AUUAGUAAAA | unstable protein |
| SUMO-RRM | R428E | RNA: AUUAGUAAAA | 2.87 ± 0.85 µM |
| SUMO-RRM | F445A | RNA: AUUAGUAAAA | >1,000 µM |
| SUMO-RRM | K447D/A | RNA: AUUAGUAAAA | unstable protein |
| SUMO-RRM | F449A | RNA: AUUAGUAAAA | 1.99 ± 0.19 µM |
| SUMO-RRM | R450A | RNA: AUUAGUAAAA | 1.91 ± 0.71 µM |
| SUMO-RRM | R472E | RNA: AUUAGUAAAA | 17.8 ± 27.5 µM |
| SUMO-RRM | F479A | RNA: AUUAGUAAAA | unstable protein |
| SUMO-RRM | D486K/A | RNA: AUUAGUAAAA | unstable protein |
| SUMO-RRM | F487A | RNA: AUUAGUAAAA | 25.6 ± 15.7 µM |
| SUMO-RRM | S492A | RNA: AUUAGUAAAA | 4.20 ± 0.56 µM |
| SUMO-RRM | R504E | RNA: AUUAGUAAAA | 2.37 ± 0.17 µM |

**Table 2 | Data collection and refinement statistics.**

| | Seb1-CID | Seb1-RRM Sulfur-SAD | Seb1-RRM Native |
|---|---|---|---|
| *Data collection* | | | |
| Space group | P 31 2 1 | C 1 2 1 | C 1 2 1 |
| Cell dimensions | | | |
| $a, b, c$ (Å) | 55.6, 55.6, 131.2 | 111.3, 47.2, 32.3 | 111.1, 47.2, 32.4 |
| $\alpha, \beta, \gamma$ (°) | 90.0, 90.0, 120.0 | 90.0, 98.9, 90.0 | 90.0, 98.9, 90.0 |
| Resolution (Å) | 48.1–1.6 (1.68–1.62)* | 55.0–2.0 (2.06–2.01) | 43.3–1.0 (1.06–1.02) |
| $R_{merge}$ | 9.8 (215.1) | 14.0 (25.9) | 4.5 (74.8) |
| $I/\sigma I$ | 14.2 (1.1) | 67.1 (8.5) | 15.5 (1.3) |
| Completeness (%) | 99.3 (98.5) | 93.3 (42.5) | 94.6 (57.5) |
| Redundancy | 8.5 (4.8) | 167.5 (15.4) | 6.0 (3.2) |
| | | | |
| *Refinement* | | | |
| Resolution (Å) | 48.1–1.6 | | 43.3–1.0 |
| No. reflections | 30,426 (2,968) | | 79,452 (5,110) |
| $R_{work}/R_{free}$ | 19.5/21.5 | | 13.5/15.2 |
| No. atoms | | | |
| Protein | 1,162 | | 1,244 |
| Ligand/ion | 0 | | 15 |
| Water | 160 | | 232 |
| *B*-factors | | | |
| Protein | 50.5 | | 13.8 |
| Ligand/ion | — | | 24.5 |
| Water | 62.5 | | 28.6 |
| R.m.s. deviations | | | |
| Bond lengths (Å) | 0.014 | | 0.010 |
| Bond angles (°) | 1.53 | | 1.48 |

SAD, single-wavelength anomalous dispersion.
*Highest resolution shell is shown in parenthesis.

Seb1 mutations drastically affects gene expression genome-wide, influencing both protein-coding and ncRNAs.

Comparison of read-though levels on individual genes in CID and RRM mutants revealed a very strong correlation (Supplementary Fig. 9b) suggesting that disrupting either of the two domains impairs 3′ end formation equally. Additionally, for genes that show significant read-through in the different mutants, a high degree of overlap is seen for both protein-coding and

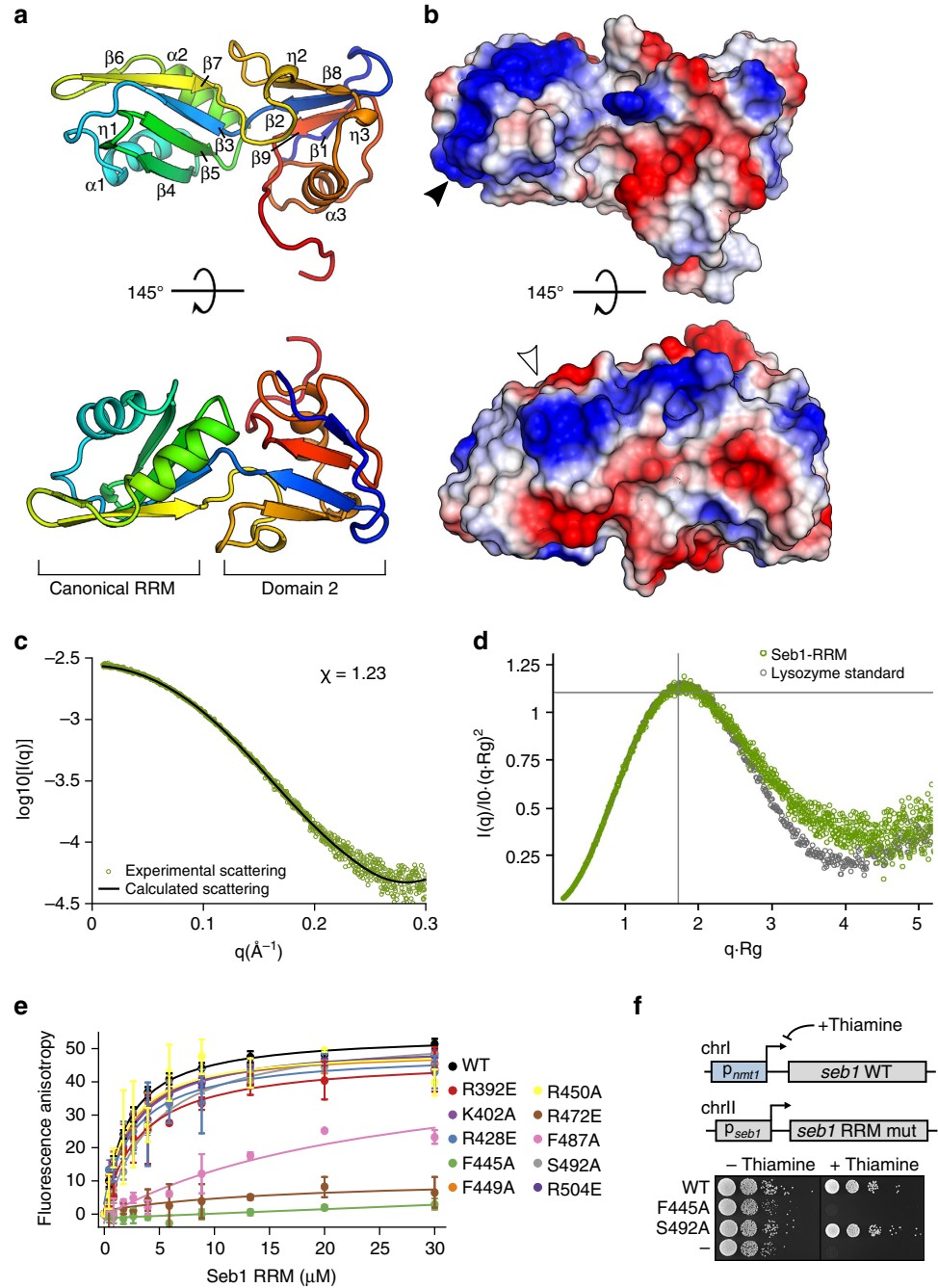

**Figure 3 | The Seb1-RRM domain has an unusual structure and RNA binding is essential.** (**a**) Crystal structure of the Seb1-RRM$_{388-540}$ domain is shown as a cartoon representation, coloured in blue to red from the N- to the C-terminus. The position of the canonical RRM and the additional RRM-like domain is indicated below the structure. The two domains are interwoven and cross over between β2 and β3. (**b**) Electrostatic surface of the Seb1-RRM$_{388-540}$ shown in the same orientations as in **a**. Positively charged areas are coloured in blue and negatively charged areas in red. Arrows indicate a contiguous positively charged region tentatively assigned to interaction with the RNA phosphate backbone. (**c**) Plot showing a solution SAXS curve of the Seb1-RRM$_{388-540}$ (green). To compare the solution and crystallographic conformations of the Seb1-RRM$_{388-540}$, a scattering profile was computed from the X-ray structure (black) and fitted to the solution scattering data. The quality of the fit as expressed as χ is indicated. (**d**) Flexibility analysis of the Seb1-RRM$_{388-540}$ (green) and a lysozyme standard (grey, BioisisID: LYSOZP) via dimensionless Kratky plot is shown. The intersection of the lines indicates the Guinier–Kratky point ($\sqrt{3}$, 1.104), the peak position of an ideal globular and rigid protein. Rigid proteins show a characteristic parabolic shape with a peak at the indicated position (as is the case here), while unfolded proteins would plateau with increasing $q$-values. (**e**) Analysis of Seb1-SUMO-RRM$_{388-540}$ binding to FAM-tagged AUUAGUAAAA RNA by FA. Error bars indicate standard deviation of three technical replicates. (**f**) Spot test showing the effect of the indicated Seb1-RRM point mutations on cell growth.

non-coding genes (Fig. 4f). We conclude that the CID and RRM domains are both required to ensure proper transcription termination. Interestingly, there is a negative correlation between changes in read-through in the mutants and basal levels of read-through in WT (Supplementary Fig. 9c), suggesting that Seb1-dependent termination is particularly efficient.

In order to assess whether loss of Seb1 recruitment is responsible for the mutant phenotypes, we compared transcriptional read-through in the mutants with recruitment of Seb1. For both ChIP-Seq and PAR-CLIP we found a very good correlation with read-through (Supplementary Fig. 9d). We split all genes into two groups, those that show Seb1 crosslinking at the PAS by

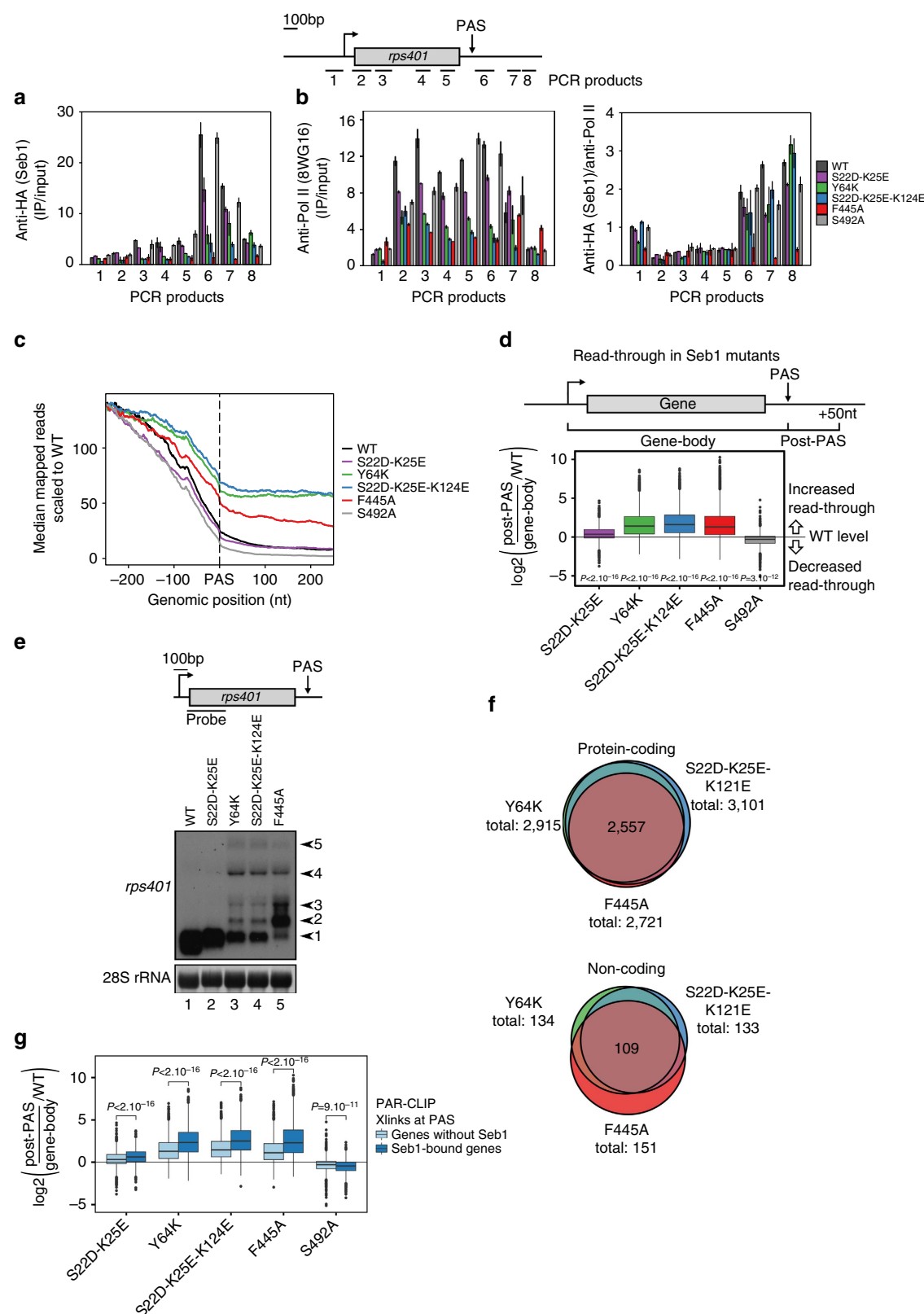

PAR-CLIP, and those that do not (Fig. 4g). Indeed, genes that bind Seb1 in WT have significantly higher levels of read-through in the mutants. This is also the case for recruitment determined by ChIP-Seq (Supplementary Fig. 9e), suggesting that the interaction between RRM-RNA (detected by PAR-CLIP), as well as binding to the transcription machinery (detected by ChIP), are both important to ensure correct transcription termination. In contrast, recruitment of Seb1 to the TSS has no influence on read-though levels (Supplementary Fig. 9f).

**Seb1 acts with Rat1/Dhp1 to terminate Pol II transcription.** Finally, we wanted to understand whether Pcf11, another CID-protein that plays a key role in transcription termination, is required at the same genes as Seb1 or if they are recruited to different subsets of genes. Similarly to Seb1, Pcf11 is known to interact with CPF components and to bind to S2P-Pol II in budding yeast[31]. However, unlike Seb1, Pcf11 relies on other components of the CPF complex for RNA recognition[43]. To compare recruitment profiles, we performed ChIP-Seq analysis of Pcf11-TAP and S2P-Pol II (Fig. 5a). As predicted, both Pcf11 and Seb1 co-localize with S2P-Pol II (Fig. 5a and Supplementary Fig. 10a–c). The overlap between genes that are bound by Pcf11 and Seb1 is remarkable (Fig. 5b and Supplementary Data 1), with 93% of genes that bind Pcf11 also binding Seb1. This suggests that Pcf11 and Seb1 are both required for proper Pol II termination but are likely to have distinct roles since both proteins are essential[44].

To examine the roles of Seb1 and Pcf11 on PAS-independent genes, we assessed their recruitment to sn/snoRNA (Fig. 5c). Here, both Seb1 and Pcf11 are also simultaneously bound. Furthermore, mutating either the CID or RRM domain of Seb1 leads to read-through transcription at snoRNAs. This suggests that Pol II relies on Seb1 as well as Pcf11 for termination at these genes. To our surprise, S2P-CTD levels peaked downstream of sn/snoRNAs, at the same position as Seb1 and Pcf11. This is in contrast to *S. cerevisiae*, where the Seb1 homologue Nrd1 is recruited to these genes via S5 phosphorylation.

To gain further mechanistic insights into the function of Seb1 we wanted to examine whether it acts in the same or a parallel pathway as the exonuclease Dhp1 which, according to the 'torpedo model', can only be recruited after a cleaved 5′ end has been generated. Therefore, we combined the *dhp1-154* mutation with mutations in *seb1*. Interestingly, the double mutants show synthetic growth defects and a more severe defect in 3′ end formation compared to each single mutant (Fig. 5d and e, lanes 4–6). Similarly, an additive effect on growth and 3′ end formation

(Fig. 5d and e, lanes 7–9) is also observed in double mutants of *seb1* and *pfs2*, which is an integral component of the CPF complex[45]. This suggests that Seb1 can also contribute to termination independently of transcript cleavage and Dhp1 (Fig. 6a).

## Discussion

During elongation Pol II forms a remarkably stable complex with the DNA template that requires several essential CID-proteins for termination. Based on studies in budding yeast, PAS-containing protein-coding and PAS-lacking non-coding genes are believed to employ different transcription termination mechanisms. Where termination occurs independently of a PAS, the CID-RRM protein Nrd1 is required[14,20]. CID-RRM proteins do not seem to be necessary for PAS-dependent termination in *S. cerevisiae*. In contrast to this widely accepted paradigm, we demonstrate that the fission yeast CID-RRM protein Seb1 drives transcription termination of both types of Pol II transcribed genes (Fig. 6b). These findings suggest that PAS-dependent and PAS-independent transcripts can utilize the same mechanism of transcription termination.

The CPF complex is recruited during transcription to mediate transcript cleavage which, in turn, is needed for transcription termination. Two models for coupling mRNA 3′ processing to termination have been proposed[46]. According to the 'allosteric model', the polymerase undergoes conformational changes upon recognition of the PAS, leading to termination[47–50]. The 'torpedo model' proposes that Rat1/Dhp1 degrades cleaved RNA until it reaches the polymerase and causes it to terminate[8]. In this model, cleavage must take place before termination can happen. This model has been challenged by reports that termination can occur without transcript cleavage[51–53]. Additionally, the Pcf11-CID can dismantle Pol II complexes *in vitro* independently of cleavage at the PAS[50]. Based on the data presented, we propose that Rat1/Dhp1 also requires assistance of CID-RRM proteins to remove Pol II from DNA.

Although Seb1 recapitulates some functions of Nrd1 (such as termination of non-coding transcripts), it also drives termination of PAS-dependent protein-coding genes and therefore plays a much more general role. Many transcripts that are dependent on Nrd1 for termination are subsequently targeted by the exosome for degradation or 3′ end trimming. In contrast to Nrd1, however, which directly recruits the nuclear exosome[26], Seb1 does not co-purify with the RNA degradation machinery. It is not clear how the exosome is targeted to transcripts for 3′ end trimming in *S. pombe*. In budding yeast, 3′ end formation of

---

**Figure 4 | Seb1 point mutations cause transcriptional read-through genome-wide. (a)** Analysis of Seb1-HA recruitment to the *rps401* gene by ChIP-qPCR. Positions of primers used are shown in the schematics above. Seb1-WT was depleted in thiamine-containing media for 24 h. Error bars indicate the standard error of biological duplicates. **(b)** Left: same as **a** but a phosphorylation-independent antibody against the Pol II-CTD was used (8WG16). Right: Same as **a** but signal was normalized to Pol II levels (shown on the left). **(c)** Median mapped reads determined by RNA-Seq in the indicated point mutants were centred on the PAS. All curves are normalized to the same starting value using the same subset of genes as in Fig. 1c. **(d)** Read-through of the different point mutants was determined by dividing mapped reads in the window PAS ± 50 nt by the read count within the gene-body (*n* = 5,119). The log2 fold change in read-through as compared to WT is shown. The significance of the overall difference between WT and each mutant was determined by the Wilcoxon–Mann–Whitney test and is indicated below each box. **(e)** Northern blot showing different transcripts derived from the *rps401* gene in the indicated mutants (cells were grown as in **a**). Arrows on the right mark individual transcripts and the position of the probe used relative to the gene is indicated in the schematics above. **(f)** Venn diagram depicting the overlap between genes that show significantly (*P* < 0.05) more read-through than WT calculated as in **d** and determined by the Kruskal–Wallis test for the indicated strains. Protein-coding (top, *n* = 4,105) and non-coding (bottom, *n* = 1,013) genes are shown separately. No genes could be found in the strains S22D-K25E and S492A that have significantly more read-through than WT. **(g)** The log2 fold read-through was calculated as in **d** for the same subset of genes as in Fig. 1c but here, all genes were split into two groups, those containing crosslinks detectable by PAR-CLIP at 250 nt ± PAS (*n* = 1,536) and those that do not (*n* = 2,692). The significance of the difference between the two groups was calculated for each mutant as in **d**. In box plots in this figure, the centre line is the median, the box limits are from the second to the third quartile (so 25% to 75% of the data points), and the whiskers extend from there to the min and max values, with outliers indicated by dots outside the whiskers.

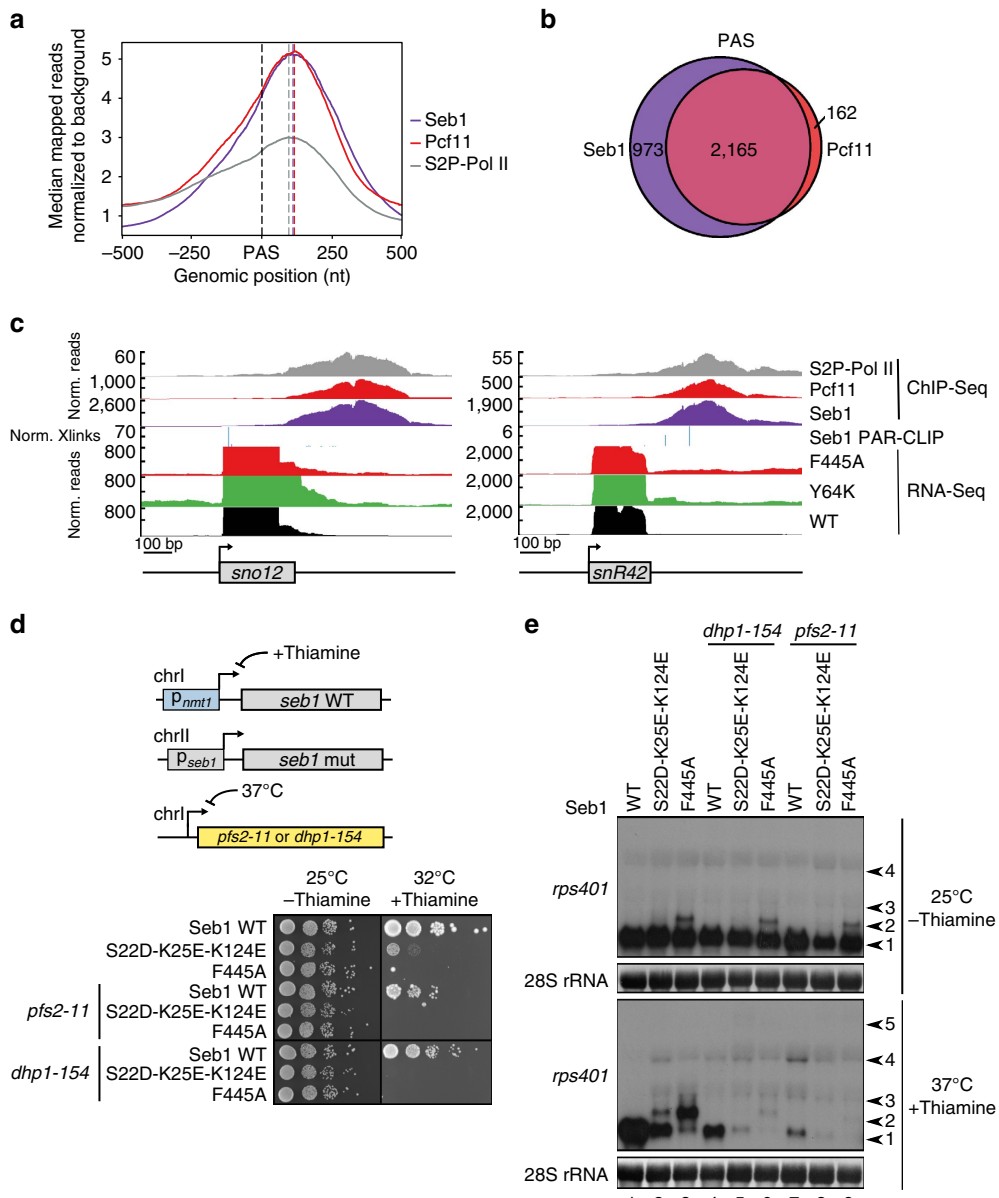

**Figure 5 | Seb1 and Pcf11 are recruited to the same genes. (a)** Averaged occupancy profiles of Seb1-TAP, Pcf11-TAP and S2P-Pol II on protein-coding genes as determined by ChIP-seq was calculated and centred at the PAS. Genes with a distance less than 500 nt to their downstream gene were excluded ($n = 2,811$). **(b)** Venn diagram depicting the overlap between genes that are bound by Seb1 and those that are bound by Pcf11 at PAS ± 250 nt. The summits of the ChIP-seq peaks were used to define binding in this window, the same subset of genes was used as in **a**. **(c)** Profiles of mapped reads normalized to *adh1* as determined by RNA-Seq of WT, Y64K and F445A after 24 h in thiamine-containing media, crosslinking sites normalized to transcript abundance from Seb1 PAR-CLIP and mapped reads normalized to a background control from ChIP-Seq of Seb1-TAP, Pcf11-TAP and S2P-Pol II are shown for two sn/snoRNA genes as indicated. **(d)** Spot test showing the effect of the indicated Seb1 point mutations in combination with either of the temperature-sensitive alleles *dhp1-154* or *pfs2-11*, as indicated, on cell growth. **(e)** Northern blot showing different transcripts derived from the *rps401* gene in the same strains as shown in **d**. Cells were grown in thiamine-containing medium for 24 h at 25 °C and shifted to 37 °C for the last 3 h before collection. The same probe was used as in Fig. 4e.

sn/snoRNAs does not require the CPF or endonucleolytic cleavage. However, this may not be the case in fission yeast. Furthermore, a connection between some components of the CPF and the exosome has been proposed[54–56].

In humans it is the Integrator complex that couples 3′ end processing of snRNAs with termination in a process that involves the negative elongation factor NELF[57]. NELF also regulates elongation through release of paused Pol II from promoters. In contrast to Nrd1-dependent termination, the endonucleolytic subunit of Integrator, INT11, cleaves pre-snRNAs co-transcriptionally[58]. Additionally, recent studies implicate Integrator in assisting NELF in regulating protein-coding genes[59,60]. Promoter-proximal transcription termination also plays a role in restricting non-coding transcription from bidirectional promoters in mammalian cells[61–65]. Interestingly, in addition to binding downstream of the PAS, we detect Seb1 near the TSS, possibly due to binding of promoter-proximal transcripts which are highly unstable in other species[65,66]. Although their origin is not clear, it is tempting to speculate that they are produced by early transcription termination similar to

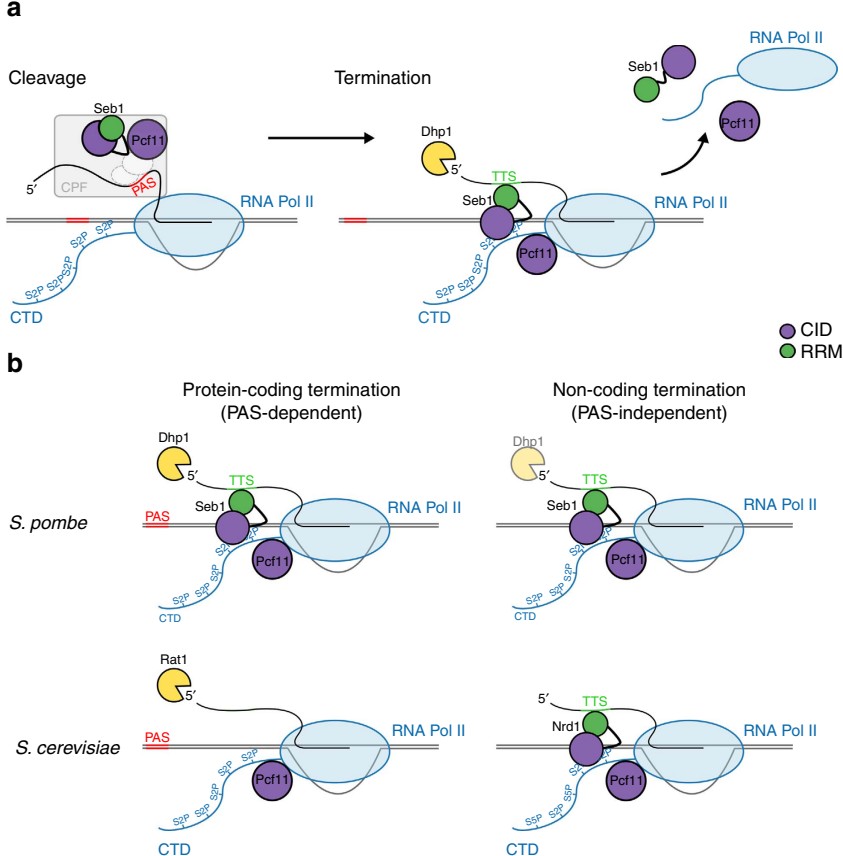

**Figure 6 | Model for the function of Seb1 and Pcf11 in transcription termination.** (**a**) A proposed model for 3′ end formation and transcription termination that relies on the cooperative action of the CID-proteins Seb1 and Pcf11. Seb1 and Pcf11 are recruited to the 3′ ends through interaction with components of the CPF complex. During transcript cleavage by the CPF/CF, Pol II is phosphorylated on S2 which allows direct CTD interaction of Seb1 and Pcf11. In addition, the Seb1 binding motif, which serves as a termination signal (TTS), is transcribed allowing for Seb1 interaction with the nascent transcript via its RRM domain. Seb1 and Pcf11, probably together with the exonuclease Dhp1, lead to the disassembly of the Pol II complex from the DNA template. (**b**) Fission yeast utilize a conserved mechanism for termination of the PAS-dependent (protein-coding) and PAS-independent (non-coding) genes. This is in contrast to budding yeast which relies on two different mechanisms for termination. In fission yeast, Seb1 and Pcf11 are necessary for termination of Pol II on both classes of genes. In budding yeast, protein-coding genes are terminated by Pcf11 and Rat1, whereas non-coding transcripts are terminated by Nrd1. Pcf11 is involved in termination of all Pol II transcribed genes in both yeasts.

Nrd1-dependent CUTs in budding yeast[29]. Consistent with this idea we find that promoter-proximal binding of Seb1 is independent of its function at the 3′ end. Promoter-proximal transcription termination mediates promoter directionality and is regulated by an elongation checkpoint which may not be unique to higher eukaryotes, as previously thought. Indeed, recent PRO-Seq experiments have also revealed Pol II pausing in fission yeast[67].

We demonstrate that the interaction between Seb1 and RNA is integral for Seb1 recruitment to chromatin, PAS cleavage, transcription termination and cell viability. The high-resolution structure of the Seb1-RRM domain presented here reveals that Seb1 utilizes an unusual structural organization composed of a canonical RRM and an additional domain for RNA binding. The configuration of the Seb1-RNA-binding module is very different from the much more flexible Nrd1 solution structure[38]. Our SAXS data confirms that the Seb1-RRM domain shows the same overall fold in solution as is observed in the structure. NMR studies of the Nrd1-RRM demonstrate chemical shifts of aa in both parts of the domain upon the addition of RNA. This, together with the high degree of conservation between the RRM domains of both proteins, may suggest that Nrd1 adopts a similar fold to Seb1 upon RNA binding. Furthermore, given that the

RNA-binding module is crucial for protein function *in vivo* and based on multiple sequence alignments (Supplementary Fig. 5a), the structural arrangement of the RNA-binding module may be conserved in other CID-RRM proteins as well.

## Methods

**Yeast strains and manipulations.** Unless indicated otherwise, *S. pombe* strains were grown in YES to $OD_{600}$ 0.4–0.7. Strains that contained the WT *seb1* gene under control of the repressible *nmt1* promoter were first grown in EMMG (−thiamine) and then shifted to YES (+thiamine) for 24 h until collection at $OD_{600}$ ∼0.5. Standard PCR-based methodology was used for epitope tagging[68]. Strains, oligonucleotides and plasmids are listed in Supplementary Tables 1–3.

**Northern blotting.** Northern blot experiments were essentially performed as described previously[26,56]. RNA was prepared with the hot phenol method and 10 µg per lane resolved on 1.2% agarose gels containing 6.7% formaldehyde in MOPS buffer. After capillary transfer in $10 \times$ SSC onto Hybond N+ membranes (GE Healthcare), RNA was UV-crosslinked and stained with methylene blue. Gene-specific probes were generated by random priming in the presence of ATP[$\alpha$-$^{32}$P] using the Prime-It II Random Primer Labeling Kit (Agilent) and hybridized at 42 °C overnight. After repeated washes in $2 \times$ SSC, 0.1 % SDS, blots were exposed on Amersham Hyperfilm MP (GE Healthcare) or quantified with a Fla-7000 phosphoimager (Fujifilm). For *adh1*, strand-specific, digoxigenin (DIG)-labelled probes were used which were generated by *in vitro* transcription with the MAXIscript kit (Ambion) and detected using the DIG system

(Roche). Uncropped versions of the northern blots are shown in Supplementary Figs 11 and 12.

**Purification of Seb1-TAP.** TAP-tagged Seb1 was purified from 16 l of yeast culture grown in YES to $OD_{600} = 0.9$. The collected cells were washed with TMN lysis buffer (20 mM Tris-HCl (pH 8.0), 5 mM $MgCl_2$, 150 mM NaCl, 10% glycerol) supplemented with 1 mM phenylmethylsulfonyl fluoride, 1 mM benzamidine, 2.3 μM leupeptin (1 μg ml$^{-1}$), 1.5 μM pepstatin A (1 μg ml$^{-1}$), 81 μM bestatin (25 μg ml$^{-1}$) and 1.5 μM aprotinin (10 μg ml$^{-1}$). Cells were resuspended in TMN and frozen into beads by dripping into liquid nitrogen. To prepare extract, yeast beads were broken and ground into a fine powder using a pestle and mortar and subsequently vortexed with glass beads in TMN buffer. The extract was centrifuged at 2,500g for 7 min, to remove glass beads, followed by ultracentrifugation at 75,000g for 1.5 h. Seb1-TAP was incubated with 1,000 μl IgG sepharose (VWR) for 16 h. Beads were washed twice with TMN buffer (plus protease inhibitors as above), thrice with TMN (without protease inhibitors) and once with TEV cleavage buffer (20 mM Tris-HCl (pH 8.0), 150 mM NaCl, 0.5 mM DTT, 0.05% NP-40, 5% glycerol). AcTEV protease (20 μl) (Invitrogen) was then added for overnight cleavage. Seb1-TAP cleavage and purification was analysed by western blot (see below) and silver stain using the SilverQuest Silver Staining Kit (Invitrogen).

**Expression and purification of recombinant proteins.** Full-length Seb1 was cloned into pET41a(+), resulting in a C-terminal His$_8$-tag. The CID point mutations were introduced by site-directed mutagenesis using primers containing the indicated mutations. The proteins were expressed in Rosetta *Escherichia coli* strain and collected by centrifugation at 4 °C and 6,200g for 15 min. Pellets were resuspended in lysis buffer (50 mM Tris-HCl, 5 mM imidazole, pH 8.0), and following lysis by French Press 1 M NaCl, 0.1% NP-40 and phenylmethylsulfonyl fluoride were added. Lysates were centrifuged at 4 °C, 100,000g for 1 h, and the supernatant was loaded onto a Ni-NTA (nickel-nitrilotriacetic acid) column. Proteins were eluted with 40–500 mM imidazole and eluted protein was used for peptide binding assays.

The CID$_{1-152}$ construct was generated from pET41a(+)-Seb1. The protein was expressed and purified as above. After elution from NiNTA, protein-containing fractions were combined and subjected to size-exclusion chromatography using a HiLoad 16/60 Superdex 200 prep grade column (GE Healthcare) on an ÄKTA purifier (GE Healthcare) in 25 mM Tris pH 7.5, 300 mM NaCl and 5 mM beta-mercaptoethanol. The protein solution was concentrated to ~4 mg ml$^{-1}$ for FA assays and ~10 mg ml$^{-1}$ for crystallization trials.

Recombinant WT and mutated RRM$_{388-540}$ of Seb1 with an N-terminal His$_6$-SUMO tag were expressed from modified pOPINS3C expression vector[69] in pLysS *E. coli* strain. For FA assays, protein eluted from NiNTA was further purified on a HiLoad 16/60 Superdex 200 prep grade column (GE Healthcare) using an ÄKTA purifier (GE Healthcare) in 50 mM Tris pH 8.0 and 200 mM NaCl, and fractions containing the Seb1-RRM$_{388-540}$ concentrated to ~5 mg ml$^{-1}$. For crystallization and SAXS, the NiNTA eluted fractions were buffer exchanged into 50 mM Tris pH 7.5, 300 mM NaCl and 500 mM NDSB-201 using a PD-10 column (GE Healthcare) and cleaved with 3C protease (Sigma-Aldrich) at room temperature overnight. His$_6$-SUMO tag was removed using NiNTA beads and the cleaved RRM$_{388-540}$ was further purified on a Superdex 75 10/100 GL column (GE Healthcare) in 20 mM HEPES pH 7.3, 150 mM NaCl and 500 mM NDSB-201. For crystallization trials, the eluted protein was concentrated to ~8.7 mg ml$^{-1}$, for SAXS to ~5 mg ml$^{-1}$.

**Western blotting.** After SDS−PAGE, proteins were transferred onto a polyvinylidene fluoride membrane via wet transfer at 30 V and 4 °C for 16 h using 25 mM Tris, 192 mM glycine, 1 mM EDTA and 20% (v/v) methanol as transfer buffer. The membrane was blocked with TBST (50 mM Tris-HCl pH 7.5, 150 mM NaCl, 0.1% Tween-20) + 5% (w/v) skim milk powder for 30 min at RT. A primary antibody (mouse α-HA (12CA5, gift from Michael Keogh) diluted 1:1,000; mouse α-Rpb1 (8WG16, Millipore cat number: 05-952) diluted 1:2,000; rabbit α-S2P Pol II (polyclonal, abcam cat number: ab5095) diluted 1:1,000; rat α-S5P Pol II (3E8, Millipore cat number: 04-1572-I) diluted 1:2,000; or rat α-S7P Pol II (4E1, Millipore cat number: 04-1570-I) diluted 1:2,000) or a horse radish peroxidase (HRP)-coupled antibody (mouse α-FLAG M2-HRP (Sigma-Aldrich cat number: A8592) diluted 1:1,000 or rabbit peroxidase α-peroxidase (PAP, Sigma-Aldrich cat number: P1291) diluted 1:2,000) were diluted in TBST + 5% skim milk powder and incubated with the membrane overnight at 4 °C. The membrane was then washed three times for 10 min with TBST. If necessary, the membrane was incubated with a secondary antibody (goat α-mouse-HRP (Sigma-Aldrich cat number: A2304), goat α-rabbit-HRP (Sigma-Aldrich cat number A0545) or goat α-rat-HRP (Calbiochem cat number: DC01L)) diluted 1:10,000 as described above and the washing was repeated as before. Proteins were subsequently visualized using Clarity Western ECL (Bio-Rad) according to the manufacturer's instructions. Uncropped versions of the western blots are shown in Supplementary Figs 11 and 12.

**Peptide binding assay.** Peptide binding assays were performed with biotinylated four-repeat CTD peptides (Peptides&elephants, Potsdam, Germany) essentially as described[13]. For this, 5 μg of peptides dissolved in TBE (10 mM Tris-HCl pH 8.0, 1 mM EDTA) were bound to 60 μl of Streptavidin-coated Dynabeads (Invitrogen) and washed thrice with OBB (25 mM Tris-HCl pH 8.0, 50 mM NaCl, 1 mM DTT, 0.03% Triton X-100, 5% glycerol). The beads were then incubated either with Seb1 purified from yeast (500 μg) or with recombinant protein (9.5 μg) and incubated at 4 °C overnight on a rotator. The beads were washed five times with OBB and the protein was eluted by boiling in SDS loading buffer and analysed by SDS–PAGE.

**Fluorescence anisotropy (FA) assay.** For Seb1-CID$_{1-152}$, binding to 75 nM of two-repeat CTD peptides containing an N-terminal 5′-fluorescein amidite (FAM)-tag (Peptides&elephants, Potsdam, Germany) was determined in 25 mM HEPES pH 7.3, 200 mM NaCl and 1 mM EDTA. For RRM$_{388-540}$, binding was determined to 40 nM FAM-AUUAGUAAAA RNA (Eurofins) in 25 mM Tris-HCl pH 8.0, 100 mM NaCl, 2 mM $MgCl_2$, 1 mM DTT, 16.7% (v/v) glycerol, 0.1% IGEPAL, 0.1 mg ml$^{-1}$ tRNA and 2.5% (v/v) RNasin (Promega). Excitation of the ligand was performed with linearly polarized light at 485 nm and emission was measured at 520 nm in parallel and perpendicular planes to the emission plane at 25 °C using a FLUOstar-Omega microplate reader (BMG-Labtech). All measurements were performed in at least duplicates, results were plotted against the protein concentration and $K_d$ values were determined via curve fitting as described in ref. 70.

**Structure determination.** Details about the crystallization experiment, data collection and processing can be found in the Supplementary Methods.

**Small angle X-ray scattering (SAXS).** Experimental details for SAXS are described in the Supplementary Methods.

**RT-qPCR.** Two micrograms of total RNA was extracted by the hot phenol method as described in ref. 26 and was subsequently digested with 2U of DNase RQ1 (Promega) for 1 h at 37 °C. RNA (100 ng) was used for reverse transcription in a total volume of 25 μl. Five microlitres were used to perform qPCR in duplicates.

**Chromatin immunoprecipitation (ChIP).** Exponentially growing cells (200 ml) were crosslinked with 11% formaldehyde solution for 20 min at room temperature. Thirty millilitres of a solution of 3M glycine, 20 mM Tris was used to quench the reaction. Cells were pelleted and washed once with cold TBS and once with FA lysis buffer (50 mM Hepes-KOH pH 7.5, 150 mM NaCl, 1 mM EDTA, 1% Triton X-100, 0.1% Na Deoxycholate)/0.1% SDS. To prepare chromatin, cells were resuspended in FA lysis buffer with 0.5% SDS and vortexed for 30 cycles of 1 min vortexing and 1 min on ice. The lysate was ultracentrifuged (150,000g, 20 min) and the pellet crushed in lysis buffer. Samples were sheared for 80 min with a sonication cycle of 15 s ON/45 s OFF with a Biorupter sonicator, and ultracentrifuged (150,000g, 20 min) to yield sheared chromatin in the supernatant. At this point the concentration of NaCl was adjusted to 275 mM. Immunoprecipitations (IPs) were conducted with 15 μl of packed rabbit IgG agarose (Sigma) or 5 μl of antibodies recognizing either HA (12CA5, gift from Michael Keogh) or Rpb1 (8WG16, Millipore cat number: 05-952) coupled to 20 μl of protein-G dynabeads (Life Technologies). After washing and eluting bound material from the beads, protein was removed by incubation with 0.2 mg pronase for 1 h at 42 °C, followed by overnight incubation at 65 °C. After phenol-chloroform extracting DNA, the amount of IP DNA relative to an input sample was determined by quantitative PCR analysis using SensiMix SYBR (Bioline).

**ChIP-Seq.** Chromatin was prepared as above from 3 × 200 ml of culture per sample. Seb1-TAP, Pcf11-TAP and S2P Pol II were IP-ed with either 15 μl of IgG agarose (Sigma-Aldrich) or 20 μl of Protein G Dynabeads (Invitrogen) with 5 μl of pre-bound α-S2P antibody (3E10, Millipore cat number: 04-1571-I). After washing and eluting bound material from the beads, three independent IPs were pooled per sample. Protein was removed by incubation with 0.4 mg pronase as described above. RNA was degraded by incubating samples with 0.02 mg RNase A (Roche) for 1 h at 37 °C. DNA was then purified using ChIP DNA Clean & Concentrator kit (Zymo Research, USA) according to the manufacturer's instructions. A sequencing library was constructed using NEBNext Fast DNA Library Prep Set for Ion Torrent Kit (NEB, USA). Libraries with different barcodes were pooled together and loaded onto the Ion PI Chip v3 using the Ion Chef Instrument (Life Technologies, USA). Library sequencing was carried out on the Ion Torrent Proton.

**PAR-CLIP.** The PAR-CLIP experiment and data analysis were performed as follows which is essentially as described[71] with minor modifications. Cells were grown at 30 °C to $OD_{600}$ ~0.5 in CSM minimal medium (Formedium) supplemented with 10 mg l$^{-1}$ uracil, 100 μM 4-thiouracil and 4% glucose. At $OD_{600}$ ~0.5, another 900 μM 4-thiouracil were added and cells were grown further for 4 h ($OD_{600}$ ~1.3–1.6) and UV-irradiated (12 J cm$^{-2}$ at 365 nm).

Generated cDNA was amplified, size-selected and quantified using a 2200 TapeStation System (Agilent Technologies). Samples were sequenced on an Illumina machine (HiSeq 1500).

**Normalization for global RNA abundance.** WT cells were treated as for PAR-CLIP using the identical labelling conditions and a UV light (365 nm) energy dose of $1 \, J \, cm^{-2}$. After bead beating, total RNA was isolated by acid phenol/chloroform extraction using Roti-Phenol (Carl Roth), and purified and concentrated using the RNA Clean Concentrator-5 (Zymo Research). Purified RNA was depleted of ribosomal RNAs using Ribo-Zero rRNA removal kit (Epicenter). The resulting rRNA-depleted RNA was used for multiplexed RNA-Seq library preparation using the NuGEN Encore Complete RNA-Seq Library Systems. Libraries were qualified on an Agilent Bioanalyzer 2100 (Agilent Technologies) and sequenced on an Illumina HiSeq 1500.

**RNA-Seq.** The indicated strains were grown in EMMG and shifted to YES media for 24 h until $OD_{600} \sim 0.5$ was reached. Cells were collected and total RNA was extracted and DNase digested as described above for RT–qPCR. After ribodepletion using the Ribo-Zero Magnetic Kit for yeast (Epicentre), libraries were generated according to the TruSeq protocol (Illumina) to generate strand-specific, second strand libraries. The sequencing was performed on an Illumina HiSeq 4000.

**Genome-wide data analysis.** Details of the data analysis conducted for ChIP-Seq, PAR-CLIP and RNA-Seq can be found in the Supplementary Methods.

**Data availability.** Raw (fastq) and processed sequencing data can be downloaded from the NCBI Gene Expression Omnibus (GEO) repository under the accession number GSE93344. The coordinates and structure factors of the Seb1-CID and Seb1-RRM have been deposited in the Protein Data Bank (PDB) under the accession codes 5MDT and 5MDU, respectively. All other data are available from the authors on reasonable request.

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

## Acknowledgements

We thank D. Hermand for the gift of strains. This work was supported by the Wellcome Trust Research Career Development and Senior Research fellowships to L.V. (WT088359MA and WT106994MA). Administrative support and support for RNA-Seq which was performed by the High-Throughput Genomics Group at the Wellcome Trust Centre for Human Genetics were supported by a Wellcome Trust Core award (090532/Z/09/Z). S.W. was supported by a studentship from the MRC and M.R. was supported by a Wellcome Trust Studentship (099667/Z/12/Z). We thank Diamond Light Source for beamtime (proposal MX8423) and the staff of beamlines I03, I04 and B21 for assistance. We also thank K. Harlos and G. Paesen for assistance with crystal manipulation.

## Author contributions

S.W. and L.V. conceived and designed the experiments. S.W. performed most of the experiments including protein purification, bioinformatics analysis, RT–qPCR and so on. C.K. helped with strain construction and northern blots. B.R.W., D.-H.H., M.H. and T.K. performed ChIPs, M.H. and T.K. contributed to bioinformatic analyses. O.A., B.R.W. and M.H. helped with protein purification and biochemical analysis. Structural analysis was done in the J.M.G. laboratory by M.R. and K.E.O. C.B. performed and analysed PAR-CLIP experiments in the P.C. laboratory. S.W. and L.V. wrote the paper and all authors edited the manuscript.

## Additional information

**Competing interests:** The authors declare no competing financial interests.

