## [Peer Review File · Nature Communications]

Reviewers' comments:

Reviewer #1 (Remarks to the Author):

Wittman and colleagues report on their studies of the Seb1 transcriptional PolII terminator from *S. pombe* using a combination of molecular biology, biochemical and structural methods. The mechanism of action of the CID-RRM proteins seems to be different depending on organism and this study enriches significantly our knowledge on the variability of the termination process. Seb1 has some distinct features from its yeast homologue Nrd1 that are convincingly supported by the data presented in the paper. The authors produced also high-resolution structures of the CID and RRM domains allowing them to establish structure function relationships. The RRM structure showed an original and unexpected topological organization.

Since the structure of the Seb1-RRM is in part different from the Nrd1-RRM NMR structure, I suggest the authors provide some quantitative information on the differences (supplementary figure with the superposition of the RRM parts etc.).

Minor remarks

Page 7

"Strikingly, in addition to expected proteins such as subunits of Pol II, a number of CPF components co-purify with Seb1 " : the identity of those proteins could be mentioned

Page 15 bottom

The sequence conservation between the RRM (especially outside the canonical RRM boxes) is not obvious to me; could the sequence similarity somehow be quantified?

There are probably some small errors in the alignment shown in FigS3A

Page 21 line 12

Figure to be referred to should be 1A rather than 1B?

Page 24 line 17

"Seb1 does not co-purify with the any RNA degradation machinery or its associated proteins" should be rephrased

Page30 lines 4-5

The structure of the Seb1-CID domain was solved using the CID domain of Nrd1 I presume ..

The access codes of the PDB files are not provided

Figure 1B

I suggest labeling of the bands in the SDS gel

Figure 3E

Some of the residues that affect RNA binding could be represented as sticks on the ribbon diagram of figure 3A

Figure 4F was missing in my pdf file

Reviewer #2 (Remarks to the Author):

This paper looks at the *S. pombe* Seb1 protein, which most closely resembles the *S. cerevisiae* Nrd1 protein. Nrd1 is essential for an alternative transcription termination pathway used at non-pA ncRNAs. In an interesting surprise, Seb1 instead functions in the normal mRNA 3' end processing pathway. The data here is very strong and convincing. There was a recently published paper (LeMay et al., *Genes Dev* 2016) that has some overlap with this story, but Wittmann and colleagues present important data that is not contained in the LeMay paper. Most notably, Wittmann et al show that Seb1 preferentially binds the serine 2-phosphorylated form of Pol II CTD, unlike the Ser5P preference of Nrd1. This essential piece of information provides a mechanism for the different targeting of the proteins. Equally important, Wittmann et al present structures for both the CTD interaction domain and RRM of Seb1 and show an interesting new variation on the RRM. Therefore, there can be no argument about novelty of the paper. I think it's an excellent candidate for *Nat Comm* and I would say it is even more appropriate for *NSMB*. However, there are multiple issues in the paper that need to be addressed before publication:

Major issues:

1. One very important question is whether Seb1 acts directly as a termination factor, or is instead needed for 3' end cleavage at the polyA site. Without cleavage, polyA factor mutants cause an indirect termination defect because the entry site for Rat1 is lost. This paper states a strong conclusion for Seb1 as a termination factor (even putting it in the paper title) and suggests there is no processing defect, but the data here actually argues the opposite. I note that the LeMay paper also has several experiments not done here also showing that Seb1 is needed for 3' cleavage. Therefore, conclusions like those found on p6 and throughout the paper about Seb1 acting as a termination factor by causing conformation changes or displacement of elongating Pol II are overstatements that are not supported by data.

a. Surprisingly, this paper doesn't actually have a direct assay for termination. In Fig 4, RNA-seq is used to monitor the effects of Seb1 mutations, but this measures steady state RNA, not nascent transcription. Very clearly there are increased reads downstream of the pA site. The text calls these transcripts downstream of the PAS "readthrough", but this isn't accurate because even in WT cells all pre-processed transcripts extend past the PAS. The RNA-seq results are perfectly consistent with a 3' processing defect. To conclude that Seb1 is directly needed for termination, the authors would need to map Pol II. They could look at nascent transcripts by Pol II PAR-CLIP or NET-seq, or do a Pol II ChIP. They would need to show that the polymerase continues transcribing past the normal termination site, which is well downstream of the PAS. This is shown by pol II ChIP in the LeMay paper.

b. One assay for defective processing is to test for RNAs that are uncleaved at the pA site. One common assay is RT-PCR with primers that bridge the pA site. If I'm reading it correctly, Fig S4E shows exactly this experiment for the *Adh1* gene and sees a big increase in uncleaved PAS. This argues strongly for a cleavage defect and I think this experiment should be in the main text. The authors could also look in their RNA-seq data for reads that cross over annotated pA sites. My prediction is there should be a very large increase genome-wide when Seb1 is depleted or mutated. Here again, the LeMay paper also reports a cleavage defect in multiple assays.

c. One other argument for Seb1 acting in processing rather than termination is the fact that it co-purifies with the pA factors but not the Dhp1 "torpedo" exonuclease or the Rai1 or Rtt103 homologs. The LeMay paper does see Dhp1 in their Seb1 preparation, so this may reflect different binding conditions and argue that Seb1 is more tightly bound to the pA factors than the termination complex. Some discussion of this should be included

2. Fig 4B I have some concern about the RNA-seq. Why do reads drop so strongly before the PAS? This suggests that the cDNA preparation step was unable to get reads near 3' ends of mature mRNAs. The methods section for the RNA-seq gives no information on how this experiment was done and whether these are random or oligo-dT primed libraries. Apparently the authors didn't finish this part of the methods, as it says "we need a citation here".

3. p14 and Fig 2H. It's surprising that the Y64K mutant doesn't kill all CTD binding due to loss of

the Tyr-Tyr interaction. The text claims Y64 is near Ser2P, but that doesn't seem to be the case in the Rtt103 co-crystal structure. What's the evidence for this? Also, the text says there's no in vivo contribution of Ser5P binding by Seb1, but clearly combining S22D-K25E with K124E gives a much stronger defect than either mutant alone. Together with the peptide binding and coIP with Ser5P, it actually appears that Seb1 may have some physiological interaction with both Ser2P and Ser5P modifications. I wouldn't rule out this possibility.

4. Given the partial overlap, it's essential to reference the recent LeMay Genes Dev paper. It's worth a paragraph to discuss the similarities and differences between the two studies. As I wrote above, the studies are complementary in many respects and the authors can cite this paper for some of the experiments that were not done here.

Minor issues:

1. The abstract mentions the RRM structure, but not the CID structure.

2. Ref 36 is correctly cited on p5 for the Rtt103/Ser2P interaction, and again on p7 for showing a role for Seb1 in heterochromatin. I think the citation for this second point should be Marina et al. Genes Dev 2013.

3. p8 There is a reference to "Watts et al., submitted" for ChIP-seq of Seb1, but the current paper also analyzes some of that data here. The Watts paper wasn't included for the reviewers to see what was in it, so it's not clear whether there's any overlap.

4. p8 What is meant by the phrase "has been retained in fission yeast"? This implies it has been lost somewhere else?

5. p9 describing the Seb1 PAR-CLIP in Fig S1D: I would clarify in the text that you mean the antisense crosslinking is upstream of the sense TSS and downstream of the sense PAS. This comes across in the figure, but was a little confusing when first reading.

6. p12, first paragraph. To my knowledge, no role for SCAF8 has been shown, so claiming the Ser2P binding of Seb1 shows homology to the human protein seems premature.

7. p22 Fig 5C - just because Seb1 and Pcf11 show ChIP to the same location, that doesn't mean they necessarily are binding simultaneously as stated. If they were competing and bound mutually exclusively they would still show overlapping crosslinking. It could be interesting to see how Seb1 depletion affects Pcf11 or pA factor binding. The LeMay paper does show that Seb1 depletion leads to reduced binding of CFI subunits, so it could be cited.

8. Fig 5C and others - It's not clear what the arrows are supposed to be pointing to. Is this to show a specific location, or are they just meant to show the general area of increased downstream reads? If the latter, I don't think you really need them.

9. Fig 6 model - There's no data in this paper regarding Y1P, so it shouldn't be included in the model. The idea that Y1P blocks pA factor binding is based on one study in *S. cerevisiae*, and as this paper shows, things in *pombe* might be different. Recent mass spec papers raise doubts about whether there is even enough Y1P in *S. cerevisiae* for this model to be true.

10. The discussion of the Rat1 torpedo model comes out overly one-sided against it (refs 50-56). However, more recent papers that strongly support the torpedo model (Pearson and Moore, JBC 2013 and Fong et al Mol Cell 2015) are not included but should be. Several of the papers cited are the old Beyer lab EM studies, where the RNA downstream of the pA site is often not seen. However, now that we know this RNA is rapidly degraded, that's not so surprising. The Bentley paper mislocalizes Xrn1 to the nucleus but finds it does not substitute for Rat1. But this is a negative result and they don't show that the Xrn1 is co-transcriptionally degrading the

downstream RNA or even interacting with the elongation complex. I think overall the evidence for the torpedo model is quite strong and I even saw a recent poster from the Cramer lab (one of the coauthors here) providing further support.

11. p26 Based on the PAR-CLIP, the authors propose that Seb1 mediates promoter-proximal termination of divergent transcripts. If that's true, one should be able to see these stabilized or extended transcripts in the RNA-seq data of Seb1 mutants. Have the authors looked at that?

12. Finally, there are a number of typos and grammar mistakes that the authors should be able to find with more careful editing. A few examples I found:

p5 "termination pathways for non-coding and protein-coding transcripts ... does not appear to be conserved" should be "do not"

p23 "employ different transcription termination mechanism.", "transcripts can utilize ubiquitous mechanism for transcription termination."

p26 "Consistently with this idea" should be "Consistent..."

p27 "nuclear exosome and TRAM complex." should be TRAMP

Figure S1 legend: "Seb1 is recruitment"

Reviewer #3 (Remarks to the Author):

The paper by Wittman et al demonstrates an important role for Seb1, the *S. pombe* homologue of *S. cerevisiae* Nrd1, in termination of transcription at genes for both non-coding and protein coding genes. This result contrasts with *S. cerevisiae* where Nrd1 function appears limited to termination at ncRNA genes that make polyA- transcripts. This wider than expected role of Seb1 in cleavage polyadenylation and termination as well as binding to UGUA like motifs was also recently reported by Lemay et al (*Genes Dev.* 30: 1558, 2016).

Specific points:

1. The conclusion in the title that "Seb1 bridges RNA polymerase II and nascent RNA" is not justified by the results. The experiments reported do not test whether bridging occurs through simultaneous occupancy of the RNA and CTD binding sites of Seb1, or alternatively whether binding to these sites is mutually exclusive.

2. The authors discuss at length the basis for the Ser2-P specificity of Seb1 but this is of limited value since the structure presented is not of a complex with a S2 phospho-CTD peptide. Indeed the physiological significance of Ser2-P specific binding is somewhat questionable given that the K121E mutant, which does not bind Ser2-P, is viable (Fig. 2G, H).

3. It would be helpful to show how Ser2-P ChIP compares with Seb1 ChIP on a few individual genes to assess how well they correlate with one another. The conclusion (p. 12) that "these data confirm that S2P is important in recruiting Seb1 to the site of transcription" is an over interpretation. Whether Seb1 recruitment requires S2 phosphorylation was not tested as far as I could see.

4. The conclusion that mutants in the CID and RRM reduce recruitment to the gene (Fig. 4A) requires a control against the trivial explanation that pol II recruitment is also reduced.

Minor point:

1. The authors should discuss whether the novel domain 2 structure formed by the sequences N and C terminal of the RRM in the fragment crystallized also forms in the context of the full length protein.

Response to the reviewers' comments

(Reviewers' comments are shown in regular font and our comments in *italics*):

We were pleased that the reviewers found our results to be significant and convincing and we appreciate the opportunity to submit a revised manuscript. The reviewers made excellent suggestions for additional improvements that we have performed. With the addition of this extra data we believe we have strengthened the main conclusions of the manuscript and hope to have satisfied all of the reviewers' concerns.

*All the genome-wide data have now been deposited and are available for reviewers to look (record **GSE89914**) following the link below while data remains in private status:*

<https://www.ncbi.nlm.nih.gov/geo/query/acc.cgi?token=slmxucwezvmdvab&acc=GSE89914>

*The structures have been deposited to PDB with the following codes **5MDT** (CID) and **5MDU** (RRM)*

Reviewers' comments:

Reviewer #1 (Remarks to the Author):

Wittman and colleagues report on their studies of the Seb1 transcriptional PolII terminator from *S. pombe* using a combination of molecular biology, biochemical and structural methods. The mechanism of action of the CID-RRM proteins seems to be different depending on organism and this study enriches significantly our knowledge on the variability of the termination process.

We thank the reviewer for the positive comments and are pleased that the reviewer found our results interesting.

Seb1 has some distinct features from its yeast homologue Nrd1 that are convincingly supported by the data presented in the paper. The authors produced also high-resolution structures of the CID and RRM domains allowing them to establish structure function relationships. The RRM structure showed an original and unexpected topological organization. Since the structure of the

Seb1-RRM is in part different from the Nrd1-RRM NMR structure, I suggest the authors provide some quantitative information on the differences (supplementary figure with the superposition of the RRM parts etc.).

As suggested, we now included a Figure comparing the Seb1 and Nrd1 RNA binding modules and the canonical RRM were aligned (Figure S4E)

Minor remarks

Page 7

“Strikingly, in addition to expected proteins such as subunits of Pol II, a number of CPF components co-purify with Seb1 “ : the identity of those proteins could be mentioned

Following the reviewer’s comment we have now added the list of proteins (Figure S1A)

Page 15 bottom

The sequence conservation between the RRM s (especially outside the canonical RRM boxes) is not obvious to me; could the sequence similarity somehow be quantified?

The alignment has been corrected and we hope that the similarity is more obvious now. Overall, the sequence identity between Seb1 and Nrd1 for the whole protein is 29.21% while the RNA binding region (Seb1-RRM₃₈₈₋₅₄₀) has 39.21% identity, which is much higher than for the protein overall. For comparison, the canonical RRM domains have an identity of 41.97% and the CID domains of 37.93%.

There are probably some small errors in the alignment shown in FigS3A

corrected

Page 21 line 12

Figure to be referred to should be 1A rather than 1B?

corrected

Page 24 line 17

“Seb1 does not co-purify with the any RNA degradation machinery or its associated proteins”
should be rephrased

Done

Page30 lines 4-5

The structure of the Seb1-CID domain was solved using the CID domain of Nrd1 I presume ..

The access codes of the PDB files are not provided

The structural data have now been deposited to the Protein Data Bank (PDB) and the entry codes are provided

Figure 1B

I suggest labeling of the bands in the SDS gel

The mass spectrometry analyis was done using entire sample therefore the identity of the bands except for Seb1 are unknown and can't be indicated

Figure 3E

Some of the residues that affect RNA binding could be represented as sticks on the ribbon diagram of figure 3A

We have now indicated mutated residues in the RRM structure (Figure S4J)

Figure 4F was missing in my pdf file

corrected

Reviewer #2 (Remarks to the Author):

This paper looks at the *S. pombe* Seb1 protein, which most closely resembles the *S. cerevisiae* Nrd1 protein. Nrd1 is essential for an alternative transcription termination pathway used at non-pA ncRNAs. In an interesting surprise, Seb1 instead functions in the normal mRNA 3' end processing pathway. The data here is very strong and convincing. There was a recently published paper (LeMay et al., Genes Dev 2016) that has some overlap with this story, but Wittmann and colleagues present important data that is not contained in the LeMay paper. Most notably, Wittmann et al show that Seb1 preferentially binds the serine 2-phosphorylated form of Pol II CTD, unlike the Ser5P preference of

Nrd1. This essential piece of information provides a mechanism for the different targeting of the proteins. Equally important, Wittmann et al present structures for both the CTD interaction domain and RRM of Seb1 and show an interesting new variation on the RRM. Therefore, there can be no argument about novelty of the paper. I think it's an excellent candidate for Nat Comm and I would say it is even more appropriate for NSMB.

We are pleased to hear that the reviewer found data presented in our manuscript solid and important.

However, there are multiple issues in the paper that need to be addressed before publication:

Major issues:

1. One very important question is whether Seb1 acts directly as a termination factor, or is instead needed for 3' end cleavage at the polyA site. Without cleavage, polyA factor mutants cause an indirect termination defect because the entry site for Rat1 is lost. This paper states a strong conclusion for Seb1 as a termination factor (even putting it in the paper title) and suggests there is no processing defect, but the data here actually argues the opposite. I note that the LeMay paper also has several experiments not done here also showing that Seb1 is needed for 3' cleavage. Therefore, conclusions like those found on p6 and throughout the paper about Seb1 acting as a termination factor by causing conformation changes or displacement of elongating Pol II are overstatements that are not supported by data.

The reviewer raised an important point. We have included data analysing cleavage efficiency at annotated PASs on individual gene (Figure 4E and S6E), as well as globally (Figure S6F).

These analyses have revealed reduced cleavage at all PASs (Figure S6F), suggesting that Seb1 is required for cleavage. As pointed out by the reviewer termination by Dhp1 is dependent on cleavage, and therefore Seb1 mutants could impact Dhp1 function and transcription termination indirectly, through defective cleavage. However, our data support that these two proteins can also act in parallel. Combining both mutations has an additive effect on growth (Figure 5D). The defect in 3' end formation observed in seb1dhp1 double mutant is significantly more severe compared to each of the single mutants (Figure 5E).

a. Surprisingly, this paper doesn't actually have a direct assay for termination.

Transcription termination is measured by Pol II ChIP (Figures 4B and 5SB)

In Fig 4, RNA-seq is used to monitor the effects of Seb1 mutations, but this measures steady state RNA, not nascent transcription. Very clearly there are increased reads downstream of the pA site. The text calls these transcripts downstream of the PAS "readthrough", but this isn't accurate because even in WT cells all pre-processed transcripts extend past the PAS. The RNA-seq results are perfectly consistent with a 3' processing defect.

We have included Northern blot analyses to complement RNA-seq showing accumulation of longer 3'extended read-through transcripts produced from rps401 (Figures 4e, 5e and S6e).

To conclude that Seb1 is directly needed for termination, the authors would need to map Pol II. They could look at nascent transcripts by Pol II PAR-CLIP or NET-seq, or do a Pol II ChIP. They would need to show that the polymerase continues transcribing past the normal termination site, which is well downstream of the PAS. This is shown by pol II ChIP in the LeMay paper.

Pol II ChIP is included (Figures 4B and 5SB)

b. One assay for defective processing is to test for RNAs that are uncleaved at the pA site. One common assay is RT-PCR with primers that bridge the pA site. If I'm reading it correctly, Fig S4E shows exactly this experiment for the Adh1 gene and sees a big increase in uncleaved PAS. This argues strongly for a cleavage defect and I think this experiment should be in the main text.

An additional analysis has been included to evaluate cleavage efficiency in seb1 mutants. This includes a Northern blot to complement the RNA-seq showing accumulation of longer 3'extended read-through transcripts produced from rps401 (Figures 4e, 5e and S6e).

The authors could also look in their RNA-seq data for reads that cross over annotated pA sites. My prediction is there should be a very large increase genome-wide when Seb1 is depleted or mutated. Here again, the LeMay paper also reports a cleavage defect in multiple assays.

As suggested by the reviewer we have analysed cleavage efficiency at all annotated PAS (Mata et al., 2013) in seb1 mutants using our RNA-seq data. Interestingly, even though the more proximal PAS are used in some cases, this analysis has revealed that cleavage is reduced at all PASs (Figure S6F),

suggesting that Seb1 is required for cleavage rather than being involved in the selection of alternative PAS as proposed by LeMay et al.

c. One other argument for Seb1 acting in processing rather than termination is the fact that it co-purifies with the pA factors but not the Dhp1 "torpedo" exonuclease or the Rai1 or Rtt103 homologs. The LeMay paper does see Dhp1 in their Seb1 preparation, so this may reflect different binding conditions and argue that Seb1 is more tightly bound to the pA factors than the termination complex. Some discussion of this should be included

We have investigated whether Seb1 acts together or in parallel to Dhp1. Since cleavage at PAS depends on Seb1, it does contribute to 'torpedo'-mediated termination. However, our data support that these two proteins can also act in parallel. Combining both mutations has an additive effect on growth (Figure 5D). The defect in 3' end formation observed in seb1 dhp1 double mutant is significantly more severe compared to each of the single mutants (Figure 5E). However, this is also true for a seb1 double mutant which pfs2, an integral part of the CPF. In some aspect Seb1 resembles Pcf11 by contributing to both cleavage and termination. As Seb1, Pcf11 also co-purifies with the CPF rather than as part of the Rat1 complex.

2. Fig 4B I have some concern about the RNA-seq. Why do reads drop so strongly before the PAS? This suggests that the cDNA preparation step was unable to get reads near 3' ends of mature mRNAs. The methods section for the RNA-seq gives no information on how this experiment was done and whether these are random or oligo-dT primed libraries. Apparently the authors didn't finish this part of the methods, as it says "we need a citation here".

The methods section has been corrected and total RNA is now clearly mentioned there and in the main text as well. The reason for the drop in reads shortly before the PAS is due to how the data was processed rather than poor library quality. The sequencing reaction was done in paired-end mode and only proper pairs (i.e. when both the forward and reverse reads map to the same chromosome in the expected direction with the expected distance in-between) were kept for analysis. Therefore, the reverse reads stemming from the pA tail would not align to the genome and the forward read would then be dropped as well, resulting in reduced reads before the PAS. However, this is only true for properly processed RNA, our read-through transcripts would still align correctly which is why we believe that this analysis is valid.

3. p14 and Fig 2H. It's surprising that the Y64K mutant doesn't kill all CTD binding due to loss of the Tyr-Tyr interaction. The text claims Y64 is near

Ser2P, but that doesn't seem to be the case in the Rtt103 co-crystal structure. What's the evidence for this? Also, the text says there's no in vivo contribution of Ser5P binding by Seb1, but clearly combining S22D-K25E with K124E gives a much stronger defect than either mutant alone. Together with the peptide binding and coIP with Ser5P, it actually appears that Seb1 may have some physiological interaction with both Ser2P and Ser5P modifications. I wouldn't rule out this possibility.

We have now investigated in more details Seb1 specificity towards CTD using FA approach (Figure 2 G-I) to complement the in vivo studies of the consequences of introducing specific point mutations on 3' end formation (Figure 4c and E) co-transcriptional recruitment of Seb1 (Figure 4A and B) and cell growth (Figure 2J). S22D-K25E-K124E where interaction with all phosphor-isoforms of Pol II is lost (Figure 2H and I) show additive effect on 3' end formation (Figures 4B and E) compared to S22D-K25E which influence S5P. S22D-K25E-K124E mutation is nearly lethal (Figure 2J) whereas S22D-K25E, K121E and K124E, which influence S5P or S2P binding, show milder growth phenotype. We conclude that even though Seb1 has highest affinity to S2P Pol II, binding to S5P is also functionally important.

4. Given the partial overlap, it's essential to reference the recent LeMay Genes Dev paper. It's worth a paragraph to discuss the similarities and differences between the two studies. As I wrote above, the studies are complementary in many respects and the authors can cite this paper for some of the experiments that were not done here.

We reference and discuss Le May et al., which had not been published at the time of initial submission of our manuscript.

Minor issues:

1. The abstract mentions the RRM structure, but not the CID structure.

CID structure is now also mentioned in the abstract

2. Ref 36 is correctly cited on p5 for the Rtt103/Ser2P interaction, and again on p7 for showing a role for Seb1 in heterochromatin. I think the citation for this second point should be Marina et al. Genes Dev 2013.

Corrected

3. p8 There is a reference to "Watts et al., submitted" for ChIP-seq of Seb1, but

the current paper also analyzes some of that data here. The Watts paper wasn't included for the reviewers to see what was in it, so it's not clear whether there's any overlap.

I believe there is no overlap between these two manuscripts, Watt et al., studies repressive role of HDAC Clr3 in a context of non-coding transcription and Seb1 is not a focus of this manuscript. Madhani's lab (Marina et al., 2014 Gene and Dev) previously proposed that these two proteins act together within heterochromatin. However, we find that Clr3 acts independently of Seb1 outside of heterochromatin.

4. p8 What is meant by the phrase "has been retained in fission yeast"? This implies it has been lost somewhere else?

Corrected

5. p9 describing the Seb1 PAR-CLIP in Fig S1D: I would clarify in the text that you mean the antisense crosslinking is upstream of the sense TSS and downstream of the sense PAS. This comes across in the figure, but was a little confusing when first reading.

Corrected

6. p12, first paragraph. To my knowledge, no role for SCAF8 has been shown, so claiming the Ser2P binding of Seb1 shows homology to the human protein seems premature.

The sentence has been removed

7. p22 FIG 5C - just because Seb1 and Pcf11 show ChIP to the same location, that doesn't mean they necessarily are binding simultaneously as stated. If they were competing and bound mutually exclusively they would still show overlapping crosslinking. It could be interesting to see how Seb1 depletion affects Pcf11 or pA factor binding. The LeMay paper does show that Seb1 depletion leads to reduced binding of CFI subunits, so it could be cited.

Indeed, as pointed out by the reviewer, we can't distinguish whether Seb1 and Pcf11 that co-localize at the 3' end of Pol II-transcribed genes bind cooperatively or compete for binding. Based on our data, we can only say that both proteins target 3' ends of a very similar set of genes and co-localize with the peak of S2P-Pol II. We now describe this section more clearly to avoid

confusion. We have also optimised Pcf11 ChIP-Seq and now Seb1 and Pcf11 datasets are more comparable (2327 transcripts bound by Pcf11 are also bound by Seb1). It would be interesting to further dissect the relationship between these essential proteins. However, the experiments suggested by the reviewer would yield data difficult to interpret due to indirect effects on recruitment due to severely altered transcription and RNA processing in the seb1 mutants. In fact, most of these experiments have been done by the LeMay et al. study which is now referenced in our manuscript.

8. Fig 5C and others - It's not clear what the arrows are supposed to be pointing to. Is this to show a specific location, or are they just meant to show the general area of increased downstream reads? If the latter, I don't think you really need them.

Arrows have been removed

9. Fig 6 model - There's no data in this paper regarding Y1P, so it shouldn't be included in the model. The idea that Y1P blocks pA factor binding is based on one study in *S. cerevisiae*, and as this paper shows, things in *pombe* might be different. Recent mass spec papers raise doubts about whether there is even enough Y1P in *S. cerevisiae* for this model to be true.

The model has been revised and Y1P removed

10. The discussion of the Rat1 torpedo model comes out overly one-sided against it (refs 50-56). However, more recent papers that strongly support the torpedo model (Pearson and Moore, JBC 2013 and Fong et al Mol Cell 2015) are not included but should be. Several of the papers cited are the old Beyer lab EM studies, where the RNA downstream of the pA site is often not seen. However, now that we know this RNA is rapidly degraded, that's not so surprising. The Bentley paper mislocalizes Xrn1 to the nucleus but finds it does not substitute for Rat1. But this is a negative result and they don't show that the Xrn1 is co-transcriptionally degrading the downstream RNA or even interacting with the elongation complex. I think overall the evidence for the torpedo model is quite strong and I even saw a recent poster from the Cramer lab (one of the coauthors here) providing further support.

Our study doesn't challenge the 'torpedo' model, in fact we now have included additional data that we believe provide an important mechanistic insights into transcription termination by demonstrating that Seb1 can act in addition to Dhp1 in transcription termination and is also required for cleavage at PAS by CPF.

11. p26 Based on the PAR-CLIP, the authors propose that Seb1 mediates promoter-proximal termination of divergent transcripts. If that's true, one should be able to see these stabilized or extended transcripts in the RNA-seq data of Seb1 mutants. Have the authors looked at that?

We believe that investigation of Seb1 role at the promoters is beyond the scope of this paper, this will provide an interesting avenue for the future studies

12. Finally, there are a number of typos and grammar mistakes that the authors should be able to find with more careful editing. A few examples I found:
p5 "termination pathways for non-coding and protein-coding transcripts ... does not appear to be conserved" should be "do not"
p23 "employ different transcription termination mechanism.", "transcripts can utilize ubiquitous mechanism for transcription termination."
p26 "Consistently with this idea" should be "Consistent..."
p27 "nuclear exosome and TRAM complex." should be TRAMP
Figure S1 legend: "Seb1 is recruitment"

All corrected

Reviewer #3 (Remarks to the Author):

The paper by Wittman et al demonstrates an important role for Seb1, the *S. pombe* homologue of *S. cerevisiae* Nrd1, in termination of transcription at genes for both non-coding and protein coding genes. This result contrasts with *S. cerevisiae* where Nrd1 function appears limited to termination at ncRNA genes that make polyA- transcripts. This wider than expected role of Seb1 in cleavage polyadenylation and termination as well as binding to UGUA like motifs was also recently reported by Lemay et al (Genes Dev. 30: 1558, 2016).

Specific points:

1. The conclusion in the title that "Seb1 bridges RNA polymerase II and nascent RNA" is not justified by the results. The experiments reported do not test whether bridging occurs through simultaneous occupancy of the RNA and CTD binding sites of Seb1, or alternatively whether binding to these sites is mutually exclusive.

We have changed the title and modified the discussion of the relevant data throughout the manuscript

2. The authors discuss at length the basis for the Ser2-P specificity of Seb1 but this is of limited value since the structure presented is not of a complex with a S2 phospho-CTD peptide. Indeed the physiological significance of Ser2-P specific binding is somewhat questionable given that the K121E mutant, which does not bind Ser2-P, is viable (Fig. 2G, H).

We have performed further analyses including the addition of the K124E mutant and quantitative binding assays of different mutants in vitro (Fig. 2g-j). We agree with the reviewer, the S2P mark does not seem to be the only determinant for binding/viability and we have changed the text accordingly.

3. It would be helpful to show how Ser2-P ChIP compares with Seb1 Chip on a few individual genes to assess how well they correlate with one another.

We have included several examples of protein-coding (Figure S8) and non-coding transcripts (Figure 5C) for S2P PolII, Seb1 and Pcf11 profiles.

The conclusion (p. 12) that “these data confirm that S2P is important in recruiting Seb1 to the site of transcription” is an over interpretation. Whether Seb1 recruitment requires S2 phosphorylation was not tested as far as I could see.

This is also brought up by the second reviewer. We have now addressed this by investigating in more details Seb1 specificity towards CTD (described above in the response to the second reviewer).

4. The conclusion that mutants in the CID and RRM reduce recruitment to the gene (Fig. 4A) requires a control against the trivial explanation that pol II recruitment is also reduced.

Seb1 ChIP data normalised to Pol II levels in corresponding strains have been included (Figure 4B) and the conclusions have been modified accordingly.

Minor point:

1. The authors should discuss whether the novel domain 2 structure formed by the sequences N and C terminal of the RRM in the fragment crystallized also forms in the context of the full length protein.

As is now mentioned in the manuscript (p. 7), the RNA binding region is flanked by intrinsically disordered regions (as predicted by DisMeta, Huang et al., Methods Mol Biol 1091:3-16 (2014)). Because of the high flexibility of the surrounding regions, the folding of the additional domain would in no way be restricted in the context of the full-length protein and would thus not interfere with the fold of the rest of Seb1.

REVIEWERS' COMMENTS:

Reviewer #1 (Remarks to the Author):

The authors have corectly taken into account most of the reviewers' remarks and I suggest that the paper now can be accepted for publication

Reviewer #2 (Remarks to the Author):

The authors have addressed all my comments.

Reviewer #3 (Remarks to the Author):

The revised manuscript properly addresses my comments and appropriate modifications have been made. In my view it is now acceptable for publication in Nature Communications.

Response to the reviewers' comments

(Reviewers' comments are shown in regular font and our comments in *italics*):

We were pleased that the reviewers are satisfied with the way we addressed their comments and we appreciated the opportunity to revise our manuscript. We would like to thank the reviewers for the review process and are very happy that they now all recommend the manuscript for publication.

REVIEWERS' COMMENTS:

Reviewer #1 (Remarks to the Author):

The authors have correctly taken into account most of the reviewers' remarks and I suggest that the paper now can be accepted for publication

Reviewer #2 (Remarks to the Author):

The authors have addressed all my comments.

Reviewer #3 (Remarks to the Author):

The revised manuscript properly addresses my comments and appropriate modifications have been made. In my view it is now acceptable for publication in Nature Communications.